# Apparent ecosystem carbon turnover time: uncertainties and robust features

Naixin Fan[1], Sujan Koirala[1], Markus Reichstein[1], Martin Thurner[3], Valerio Avitabile[4], Maurizio Santoro[5], Bernhard Ahrens[1], Ulrich Weber[1], Nuno Carvalhais[1,2]

[1]Max Planck Institute for Biogeochemistry, Hans Knöll Strasse 10, 07745 Jena, Germany

[2]Departamento de Ciências e Engenharia do Ambiente, DCEA, Faculdade de Ciências e Tecnologia, FCT, Universidade Nova de Lisboa, 2829-516 Caparica, Portugal

[3]Biodiversity and Climate Research Centre (BiK-F), Senckenberg Gesellschaft für Naturforschung, Senckenberganlage 25, 60325 Frankfurt am Main, Germany

[4]European Commission, Joint Research Centre, Via E. Fermi 2749, 21027 Ispra, Italy

[5]Gamma Remote Sensing, 3073 Gümligen, Switzerland

*Correspondence to*: Naixin Fan (nfan@bgc-jena.mpg.de) and Nuno Carvalhais (ncarval@bgc-jena.mpg.de)

**Abstract.** The turnover time of terrestrial ecosystem carbon is an emergent ecosystem property that quantifies the strength of
land surface on the global carbon cycle – climate feedback. However, observational and modelling based estimates of carbon turnover and its response to climate are still characterized by large uncertainties. In this study, by assessing the apparent whole ecosystem carbon turnover times ($\tau$) as the ratio between carbon stocks and fluxes, we provide an update of this ecosystem level diagnostic and its associated uncertainties in high spatial resolution (0.083º) using multiple, state-of-the-art, observation-based datasets of soil organic carbon stock ($C_{soil}$), vegetation biomass ($C_{veg}$) and gross primary productivity
(GPP). Using this new ensemble of data, we estimated the global median $\tau$ to be $43^{+7}_{-7}$ years ($median^{+difference\ to\ percentile\ 75}_{-difference\ to\ percentile\ 25}$) when the full soil is considered, in contrast to limiting it to 1m depth. Only considering the top 1m of soil carbon in circumpolar regions (assuming maximum active layer depth is up to 1 meter) yields a global median $\tau$ of $37^{+3}_{-6}$ years, longer than the previous estimates of $23^{+7}_{-4}$ years (Carvalhais et al., 2014). We show that the difference is mostly attributed to changes in global $C_{soil}$ estimates. $C_{soil}$ accounts for approximately 84% of the total
uncertainty in global $\tau$ estimates; and GPP also contributes significantly (15%), whereas $C_{veg}$ contributes only marginally (less than 1%) to the total uncertainty. The high uncertainty in $C_{soil}$ is reflected in the large range across state-of-the-art data products, where full-depth $C_{soil}$ spans between 3362-4792 PgC. The uncertainty is especially high in circumpolar regions, with an uncertainty of 50% and a low spatial correlation between the different datasets ($0.2 < r < 0.5$) when compared to other regions ($0.6 < r < 0.8$). These uncertainties cast shadow on current global estimates of $\tau$ in circumpolar regions, for
which further geographical representativeness and clarification on variations of $C_{soil}$ with soil depth are needed. Different GPP estimates contribute significantly to the uncertainties of $\tau$ mainly in semi-arid and arid regions, whereas $C_{veg}$ causes the

uncertainties of $\tau$ in the subtropics and tropics. In spite of the large uncertainties, our findings reveal that the latitudinal gradients of $\tau$ are consistent across different datasets and soil depths. The current results show a strong ensemble agreement on the negative correlation between $\tau$ and temperature along latitude that is stronger in temperate zones (30ºN-60ºN) than in

the subtropical and tropical zones (30ºS-30ºN). Additionally, while the strength of the $\tau$-precipitation correlation was dependent on the $C_{soil}$ data source, the latitudinal gradients also agree among different ensemble members. Overall, and despite the large variation in $\tau$, we identified robust features in the spatial patterns of $\tau$ that emerge beyond the differences stemming from the data driven estimates of $C_{soil}$, $C_{veg}$ and GPP. These robust patterns, and associated uncertainties, can be used to infer $\tau$-climate relationships and for constraining contemporaneous behavior of ESMs, which could contribute to

uncertainty reductions in future projections of the carbon cycle - climate feedback. The dataset of $\tau$ is openly available at https://doi.org/10.17871/bgitau.201911 (Fan et al., 2019).

## 1 Introduction

Terrestrial ecosystem carbon turnover time ($\tau$) is the average time that carbon atoms spend in terrestrial ecosystems from initial photosynthetic fixation until respiratory or non-respiratory loss (Bolin and Rodhe, 1973; Barrett, 2002; Carvalhais et al., 2014).

Ecosystem turnover time is an emergent property that represents the macro-scale turnover rate of terrestrial carbon that results from different processes such as plant mortality and soil decomposition. Alongside photosynthetic fixation of carbon, $\tau$ is a critical ecosystem property that co-determines the terrestrial carbon storage and the terrestrial carbon sink potential. The magnitude of $\tau$ and its sensitivity to climate change is central to modelling carbon cycle dynamics. Therefore, $\tau$ has been used as a model evaluation diagnostic and to constrain Earth system model (ESM) simulations of the carbon cycle. These analyses

have shown that current ensembles of ESMs show a large spread in the simulation of soil and vegetation carbon stocks and its spatial distribution, mostly attributed to the differences in $\tau$ among ESMs (Friend et al., 2014; Todd-Brown, 2013,2014; Wenzel, et al., 2014, Carvalhais et al. 2014; Thurner et al., 2017).

At large scales, and for ecosystem level comparisons, model simulations and observations do not agree in the global distribution of $\tau$ and its relationship with climate. Previous observational datasets, covering both lower latitudes and

circumpolar regions, used to estimate global $\tau$ for comparison with ESM simulations from the fifth phase of the Coupled Model Intercomparison Project (CMIP5) have shown a generalized tendency of the models towards faster turnover times of carbon, more sensitive to temperature when compared to observationally/based estimates (Carvalhais et al., 2014). The variability between the ESMs alone were also substantial, showing a wide range of $\tau$ from 8.5 to 22.7 years (mean difference of 29%) leading to a substantial divergence in global simulated total terrestrial carbon stocks, that range from 1101 PgC to 3374 PgC

(mean difference of 36%). The models also exhibit a large discrepancy in the $\tau$-temperature and $\tau$-precipitation relationships across different latitudes compared to observations. The difficulty of evaluating the response of soil carbon to climate change is partly due to the fact that the dynamical observations at relevant timescales, e.g. multi-decadal to centennial scales, are lacking and the magnitude of projected change of $\tau$ to climate change is still poorly constrained (Koven et al., 2017).

Current understanding of the factors that drive changes in $\tau$ are unclear due to the confounding effects of temperature and moisture even though, for instance, it is well perceived that temperature and water availability are the main climate factors that affect root respiration and microbial decomposition (Raich, J. and W. H. Schlesinger,1992; Davidson and Janssens, 2006; Jackson, R. B., et al., 2017). Therefore, it is difficult to implement local temperature sensitivity of $\tau$ into carbon cycle models due the large discrepancy between intrinsic and apparent sensitivity of $\tau$ to temperature. As the soil environment and climate are highly heterogeneous in space, the temperature sensitivity of $\tau$ and terrestrial carbon fluxes may be substantially affected by other factors as spatial scale decreases (Jung et al., 2017). Additional challenges emerge in understanding the role of climate and other environmental factors in defining vegetation dynamics related to mortality and recovery trajectories that control the plant-level contribution to $\tau$ (Friend et al., 2014; Thurner et al., 2016). Large uncertainties in the simulated total carbon stock of soil and vegetation represent process uncertainty or potentially missing processes that lead to diverse or even opposite responses of $\tau$ to changes in climate (Friedlingstein et al., 2006; Friend et al., 2014). Thus, it is instrumental to use observational-based estimations of carbon turnover times and their associated uncertainties in order to constrain the models and better predict the response of the carbon cycle to climate change.

On the other hand, the observation-based estimates of carbon turnover times themselves are prone to uncertainties stemming from the different data sources of different components of $\tau$: soil and vegetation stocks, and ecosystem carbon flux. Specifically, estimates of global total carbon stocks are characterized by large uncertainties as different in-situ measurements and upscaling methods are used to derive total carbon stocks (Batjes, 2016; Hengl et al., 2017; Sanderman et al., 2017). Alongside recent soil carbon datasets (Tifafi et al., 2018), there are also several different global vegetation biomass estimates (Thurner et al., 2014; Avitabile et al., 2016; Saatchi et al., 2017; Santoro et al., 2018) and gross primary productivity datasets (GPP, Jung et al., 2017), which may lead to substantial differences in the global $\tau$ distribution and its relationship with climate. Thus, building and evaluating an observation-based ensemble of global $\tau$ estimates derived from different products is key to quantify the uncertainties in the $\tau$-climate relationships.

This study thus aims at developing an ensemble global estimation of $\tau$ at spatial resolution of 0.083º, derived from different observation-based products. Specifically, we will (1) update $\tau$ estimations with multiple state-of-the-art datasets; (2) quantify the contribution of the different components of $\tau$ to the global and local uncertainties; (3) identify the robust patterns across the different ensemble members.

## 2 Datasets

The attributes of the $\tau$ dataset provided in this study, and the key external datasets that were used to estimate $\tau$ are summarized in Table 1. Details for each dataset are described in the following subsections. Note that all the datasets are harmonized into the same spatial resolution of 0.083º (~10km) using a mass conservative approach (see Section S1 of Supplementary Material).

### 2.1 Soil organic carbon datasets

Five different estimates of global soil carbon stock ($C_{soil}$) were obtained from independent datasets. The main features of the datasets and the approaches used are briefly described below:

a. SoilGrids is an automated soil mapping system that provides consistent spatial predictions of soil properties and types at the spatial resolution of 250 m (Hengl et al., 2017). Global compilation of in-situ soil profiles measurements is used to produce an automated soil mapping based on machine learning algorithms. The dataset contains global soil organic

carbon content at soil depths of 0, 5, 15, 30, 60, 100 and 200 cm. In addition, physical and chemical soil properties such as bulk density and carbon concentration are provided. 158 remote-sensing based covariates including land cover classes and long-term averaged surface temperature were used to train the machine learning model at the site level. According to Hengl et al. (2017), the current version of the dataset explains 68.8% of the variance in soil carbon stock compared to mere 22.9% in the previous version (22.9%) (Hengl et al., 2014). However, it has also been recognized that the

SoilGrids may overestimate carbon stocks due to high values of bulk soil density (Tifafi et al., 2018). In general, the estimation of $C_{soil}$ is mainly caused by the geographically-biased availability of measured data, especially in the circumpolar regions. Even though in-situ measurements had a large spatial extent and cover most of the continents, the remote regions that are characterized by severe climate were much less sampled.

b. The dataset of soil carbon provided by Sanderman et al. (2017, hereafter Sanderman) used the same method as SoilGrids

but different input covariates. The main difference between SoilGrids and Sanderman is that in addition to topographic, lithological and climatic covariates, Sanderman also included land use and forest fraction as covariates in the model fitting. The relative importance analysis based on the Random Forest method showed that soil depth, temperature, elevation and topography are the most important predictors of soil carbon, which is consistent with the SoilGrids. Land use types such as grazing and cropping land area also contribute significantly to the variance. The Sanderman dataset

provides soil carbon stocks for the soil depths of 0-30 cm, 30-100 cm and 100-200 cm. The dataset is available at a spatial resolution of 10 km.

c. Harmonized World Soil Database (HWSD) harmonized the soil data from more than 16000 standardized soil-mapping units worldwide into a global soil dataset (Batjes et al., 2016). It combines regional and national soil information to estimate soil properties, and yet reliability of the data varies due to the different data sources. The database derived from

the Soil and Terrain (SOFTER) database had the highest reliability (Central and Eastern Europe, the Caribbean, Latin America, Southern and Eastern Africa) while the database derived from the Soil Map of the World (North America, Australia, West Africa and Southern Asia) has a relatively lower reliability. The HWSD dataset is available at a spatial resolution of 30 arc-second, and it includes soil organic carbon and water storage capacity at topsoil (0-30 cm) and subsoil (30-100 cm).

d. The Northern Circumpolar Soil Carbon Database (NCSCD) quantifies the soil organic carbon storage in the northern circumpolar permafrost area (Hugelius et al., 2013). The dataset contains soil organic carbon content for soil depths of 0-30, 0-100, 100-200, 200-300 cm. The soil samplings included pedons from published literature, existing datasets and

unpublished material. The data for 200 and 300 cm depths were obtained by extrapolating the bulk density and carbon content values at the deepest available soil depth for a specific pedon. Only the pedons with at least the data for the first 50 cm were extrapolated to the full soil depth. The deep soil carbon (100-300 cm) showed the lowest level of confidence due to lack of in-situ measurements and much lower spatial representativeness.

e.   The soil carbon stock and properties produced by the LandGIS maps development team (hereafter LandGIS) were also used in this study (Wheeler and Hengl, 2018). The soil profiles measurements used in the training have a wide geographic coverage in America, Europe, Africa and Asia. One unique feature of LandGIS is that it includes additional soil profiles in Russia from the Dokuchaev Soil Science Institute/Ministry of Agriculture of Russia, improving the predictions of $C_{soil}$ significantly there. Further, different machine learning methods including random forest, gradient boosting and multinomial logistic regression were used to upscale the soil profiles to a global gridded dataset. Continuous soil properties were predicted at 6 different soil depths: 0, 10, 30, 60, 100 and 200cm. Compared to the SoilGrids dataset, LandGIS added new remote sensing layers as covariates in the training and used 5 times more training datasets (360000 soil profiles compared to 70000 in SoilGrids).

## 2.2 Vegetation biomass datasets

Four different datasets of biomass were used to produce the total vegetation biomass ($C_{veg}$) data at the global scale.

a.   Thurner et al. (2014) estimated the above-ground biomass (AGB) and below-ground biomass (BGB) for northern hemisphere boreal and temperate forests based on satellite radar remote sensing retrievals of growing stock volume (GSV) and field measurements of wood density and biomass allometry. The carbon stocks of tree stems were estimated from GSV retrieval of the BIOMASAR algorithm. The BIOMASAR algorithm uses remote sensing observations from the ASAR instrument on Envisat Satellite (Santoro et al., 2015), which is converted to biomass using wood density information. The other tree biomass compartments (BC) including roots, foliage and branches were estimated from stem biomass using field measurements of biomass allometry. The total carbon content of the vegetation was then derived as the sum of the biomass in different compartments and converted to carbon mass units using carbon fraction parameters. Comparison between the data with inventory-based estimates shows a good agreement at regional scales in Russia, the United States and Europe (Thurner et al., 2014). The data from Thurner et al, at 0.01° spatial resolution and representative for the year 2010, only covers the northern boreal and temperate forests between 30°N and 80°N latitudes.

b.   To accommodate for lower latitudes not covered in Thurner et al data, we used the forest biomass carbon stocks to cover the tropical regions provided by Saatchi et al., (2011). The data was derived using lidar, optical and microwave satellite imagery, trained with in-situ measurements in 4079 forest inventory plots (Saatchi et al., 2011). Using the GLAS Lidar observations to sample forest structure, the method applies a power-law functional relationship to estimate biomass from

the Lidar-derived Lorey's height of the canopy. This extended sample of biomass density is then extrapolated over the landscape using MODIS and radar imagery, resulting in a pantropical AGB map. BGB was estimated as a function of AGB and the two were summed together to derive total forest carbon stock at 1 km spatial resolution.

c. The GlobBiomass map (Santoro et al., 2018) estimated GSV and AGB density at the global scale for the year 2010 at 100 m spatial resolution. The AGB was derived from GSV using spatially explicit Biomass Expansion and Conversion Factors (BCEF) obtained from an extensive dataset of wood density and compartment biomass measurements. GSV was estimated using space-borne SAR imagery (ALOS PALSAR and Envisat ASAR), Landsat-7, ICESAT LiDAR and auxiliary datasets, using the BIOMASAR algorithm to relate SAR backscattered intensity with GSV (Santoro et al., 2018b).

d. Avitabile et al., 2016 combined two existing AGB datasets (Saatchi et al., 2011; Baccini et al., 2012) to produce data for pantropical AGB. This data uses a large independent reference biomass dataset to calibrate and optimally combine the two maps. The data fusion approach is based on the bias removal and weighted-average of the input maps, which integrates the spatial patterns of the reference data into the combined data. The resulting data of total AGB stock for the tropical regions was 9-18% lower than the two reference datasets with distinctive spatial patterns over large areas. The combined data from Avitabile et al. is available at a spatial resolution of 1 km.

## 2.3 Soil depth dataset

The data for global distribution of soil depth was obtained from the Global Soil Texture and Derived Water-Holding Capacities database (Webb, et al., 2000). The data contains standardized values of soil depth and texture selected from the values from the same soil type within each continent. The total soil depth depends on soil texture and water availability, and it is usually deeper than 100 cm. In regions with permafrost, total soil depth can extend beyond 400 cm (Figure S1).

## 2.4 Gross primary productivity datasets

The GPP datasets used to calculate ecosystem carbon turnover times were obtained from the FLUXCOM initiative (http://fluxcom.org/). In FLUXCOM, the global energy and carbon fluxes are upscaled from eddy covariance flux measurements, using different machine learning approaches with several meteorological and Earth observation data (Jung et al., 2017). In this study, we used GPP derived from the two different FLUXCOM setups, based on: (1) only remote-sensing covariates; (2) both remote-sensing and meteorology forcing (Tramontana et al., 2016; Jung et al., 2020). In this study, we derived the long-term mean annual GPP across different machine learning methods over the time period from 2001 to 2015.

## 2.5 Climate datasets

A high spatial resolution (~1 km) climate dataset WorldClim (Fick and Hijmans, 2017) was used to investigate the relationship between τ and climate. The data included monthly maximum, minimum and average temperature, precipitation, solar radiation, vapor pressure and wind speed. The WorldClim data was produced by assimilating 9000-60000 ground-station measurements and covariates such as topography, distance to the coast, and remote-sensing satellite products including maximum and minimum land surface temperature, and cloud cover in model fitting. For different regions and climate variables, different combinations of covariates were used. The two-fold cross-validation statistics showed a very high model accuracy for temperature-related variables (r > 0.99), and a moderately high accuracy for precipitation (r = 0.86).

Table 1. Overview of the data used and produced in this study.

| Dataset | Dataset abbreviation used in this manuscript | Spatial domain | Spatial resolution | Depth distribution (cm) | Original data format | Original data source |
|---|---|---|---|---|---|---|
| **C$_{soil}$** | | | | | | |
| Sanderman et al. (2017) | Sanderman | Global | 10 km | 0,30,100,200 | GeoTIFF | https://github.com/whrc/Soil-Carbon-Debt/tree/master/SOCS |
| SoilGrids | SoilGrids | Global | 250 m | 0,5,15,30,60, 100,200 | GeoTIFF | https://files.isric.org/soilgrids/data/ |
| LandGIS | LandGIS | Global | 250 m | 0,10,30,60,1 00,200 | GeoTIFF | https://zenodo.org/record/2536040#.XhxHRBf0 kUF |
| Harmonized World Soil Database | HWSD | Global | 1 km | 0,30,100 | Raster | http://www.fao.org/soils-portal/soil-survey/soil-maps-and-databases/harmonized-world-soil-database-v12/en/ |
| The Northern Circumpolar Soil Carbon Database | NCSCD | Circumpolar (30°N-80°N) | 1 km | 0,30,60,100, 200,300 | GeoTIFF/ NetCDF | https://bolin.su.se/data/ncscd/ |
| WoSIS Soil Profile Database | WoSIS | Global | In-situ | 0-300 | Shape | https://www.isric.org/explore/wosis/accessing-wosis-derived-datasets |
| International Soil Carbon Network | ISCN | Global | In-situ | 0-400 | Spreadsheet | https://iscn.fluxdata.org/ |
| Global Soil Texture And Derived Water- | Webb | Global | 100km | Not applicable | ASCII | https://daac.ornl.gov/SOILS/guides/Webb.html |

| | | | | | | |
|---|---|---|---|---|---|---|
| Holding Capacities database | | | | | | |
| **C$_{veg}$** | | | | | | |
| Global biomass dataset | Saatchi | Global | 1km | Not applicable | GeoTIFF | Dataset available through direct correspondence (Saatchi et al., 2011) |
| GEOCARBON global forest biomass | Avitabile | Global | 1km | Not applicable | GeoTIFF | http://lucid.wur.nl/datasets/high-carbon-ecosystems |
| Integrated global biomass dataset | Saatchi-Thurner | Global | 1km | Not applicable | GeoTIFF | https://www.pnas.org/content/108/24/9899 https://onlinelibrary.wiley.com/doi/full/10.1111/geb.12125 |
| GlobBiomass | Santoro | Global | 1km | Not applicable | GeoTIFF | https://globbiomass.org/ |
| **GPP** | | | | | | |
| FLUXCOM | GPP (driven by remote sensing) | Global | 10km | Not applicable | NetCDF | http://www.fluxcom.org/ |
| FLUXCOM | GPP (driven by remoting sensing + meteorology) | Global | 50km | Not applicable | NetCDF | http://www.fluxcom.org/ |
| **Climate** | | | | | | |
| WorldClim | Mean annual temperature (T) Mean annual precipitation (P) | Global | 1km | Not applicable | GeoTIFF | http://worldclim.org/version2 |
| **τ database** | | | | | | |
| BGI τ database | Terrestrial carbon turnover times | Global | 50km | 100, 200, full depth | NetCDF | https://www.bgc-jena.mpg.de/geodb/projects/FileDetails.php |

## 3 Methods

### 3.1 Estimation of ecosystem turnover times

As a result of the balance between influx and outflux of carbon, the terrestrial carbon pool can be approximated to reach the steady-state condition (influx equal outflux) when long timescales are considered. This simplifies the calculation of τ to the ratio between the total terrestrial carbon storage and the influx or the outflux of carbon. The approach is advantageous to represent the highly heterogeneous intrinsic properties of the terrestrial carbon cycle as an averaged apparent ecosystem property which is more intuitive to infer large scale sensitivity of τ to climate change. Instead of focusing on the heterogeneity

of individual compartment turnover times we show the change of carbon cycle on the ecosystem level using $\tau$ as an emergent diagnostic property. The total land carbon storage can be estimated by summing soil carbon stocks derived from extrapolation and vegetation biomass. Assuming steady state in which the total efflux (autotrophic and heterotrophic respiration, fire, etc.) equals to influx (GPP). Then $\tau$ can be calculated as the ratio between carbon stock and influx:

$$\tau = \frac{C_{soil} + C_{veg}}{GPP} \qquad (1)$$

Here $C_{soil}$ and $C_{veg}$ are the total soil and vegetation carbon stocks, respectively, and GPP is the total influx to the ecosystem An ensemble of $\tau$ estimates is generated by combining three soil carbon stocks at three different soil depths (1m, 2m, full soil depth), four vegetation biomass products, and 24 GPP, resulting in an ensemble with 864 members.

## 3.2 Estimation of global vegetation biomass stock

Two different corrections had to be addressed in order to assess the whole vegetation carbon stock from current observation-based products. First, the aboveground biomass datasets only consider the biomass within woody vegetation (mostly trees), while the biomass of herbaceous vegetation is missing. To account for herbaceous biomass, we used a previously developed method, in which the live vegetation fraction is assumed to have a mean turnover time of one year and a uniform distribution of respiratory costs of carbon (Carvalhais et al., 2014). The carbon in herbaceous vegetation can then be expressed as a function of GPP:

$$C_H = GPP \cdot (1 - \alpha) \cdot f_H \qquad (2)$$

Where $C_H$ is the carbon stock of the herbaceous vegetation; GPP is the gross primary productivity from FLUXCOM; $\alpha$ is respiration cost of carbon (0.25-0.75); and $f_H$ is the fraction of a grid cell covered by herbaceous vegetation, which was obtained from the SYNMAP database (Jung et al., 2006).

Second, two of the vegetation biomass datasets (GlobBiomass and the Avitabile, see Table 1) do not include BGB. For consistency across all $C_{veg}$ datasets, we estimated the BGB using a previously developed empirical relationship (Saatchi et al., 2011) between AGB and BGB:

$$BGB = 0.489 \cdot AGB^{0.89} \qquad (3)$$

**3.3 Extrapolation of soil datasets**

We used observed soil profiles and multiple empirical models to extrapolate soil carbon stock to full soil depth (Figure S1 and
Table S1). This approach is necessary to obtain the accumulated carbon stock from surface to full soil depth because the soil
datasets only extend up to 2 meters below the surface. However, a large amount of $C_{soil}$ is stored below this depth, especially
in peatland regions where soil carbon content can be substantially higher in deeper soil layers (Hugelius et al., 2013). To
estimate the total carbon storage in the land ecosystem, different empirical mathematical models were used (Table S1). The
Covariance Matrix Adaptation Evolution Strategy (CMA-ES) method was used to optimize parameters of the models which
is based on an evolutionary algorithm which used the pool of stochastically generated parameters of a model as the parents for
the next generation (Hansen et al., 2001).

Extrapolation using empirical numerical models may cause arbitrary bias and higher uncertainty if the models are not
appropriately chosen. Here we used the in-situ observational data from the World Soil Information Service (WOSIS) (Batjes
et al., 2019) and the International Soil Carbon Network (ISCN) (Nave et al., 2017) to select the ensemble of the models that
could best simulate soil carbon stocks at full depth. We used a global dataset of soil depth (Webb, 2000) as the maximum soil
depth that we extrapolated to. The approach performs fitting each empirical model against cumulative $C_{soil}$ with all data points
and then predicted the cumulative $C_{soil}$ at full soil depth for each soil profile independently. The ability of a particular empirical
model or combination of models was then evaluated by comparing the predictions of $C_{soil}$ at full depth against the observations
(see Supplementary Section S3.2). This procedure was applied to the two different in-situ datasets: WOSIS which covers most
of the biomes and ISCN which has more coverage in circumpolar regions. Finally, after comparing different model averaging
methods (see Supplementary Table S2) we chose two model ensembles that could best represent circumpolar and non-
circumpolar regions based on observational datasets, respectively. The performance of the chosen ensembles is synthesized in
Figure S3 and S4. Finally, each model ensemble is applied to extrapolate $C_{soil}$ to full depth in corresponding region (see
Supplementary Section S3).

**3.4 Uncertainty estimation**

To estimate the sources of uncertainty in $\tau$, we performed a N-way analysis of variance (ANOVA) on the different variables
($C_{soil}$, $C_{veg}$, and GPP). The ANOVA provides the sum of squares of each variable and the total sum of squares of all variables.
The contribution of each variable (data from different sources) to the total uncertainty can then be calculated as,

$$C_n = \frac{SS_n}{SS_{total}}$$

(4)

Where $C_n$ is the relative contribution of uncertainty from the $n^{th}$ variable, $SS_n$ is the sum of square of the $n^{th}$ variable, $SS_{total}$ is
the total sum of square of all variables. Note that the uncertainty was quantified in two domains:

1. Grid cell: The relative contributions of different variables to uncertainty in τ were calculated independently for each grid cell.

2. Global: The same method was applied to the estimate of the global τ, which is calculated using the global total carbon stocks in vegetation and soil, and GPP.

## 3.5 The analysis of zonal correlations

The local correlation between τ and climate across latitudes was obtained by using a zonal moving window approach in which the Pearson partial correlations between τ and MAT/MAP were calculated using a 360º (longitudinal span) ×2.5º (latitudinal span) moving window. This approach allowed for the assessment of the correlation strength between τ and each climate parameter. The τ values below the local 1st percentile and above the 99th percentile was removed in each moving window to avoid the effect of potential outliers in the correlations with climate. In order to investigate the effect of latitudinal span, we chose different band size of 0.5º, 2.5º and 5º and performed the correlation analysis in the same manner for each selection.

## 4 Results

### 4.1 The global carbon stock

Table 2 summarizes the estimates of $C_{soil}$ $C_{veg}$ and GPP. Globally, estimates of soil carbon stocks within the top 2-meters of soil are 2863 PgC, 3969 PgC and 3710 PgC for the datasets of Sanderman, SoilGrids and LandGIS, respectively (bulk density corrected, see Supplementary Section S2). The significant differences among different datasets indicate a high uncertainty in current estimation of global soil carbon storage. The extrapolation of $C_{soil}$ to the full soil depth (FD) shows that approximately 18% of soil carbon is stored below the depth of 2 m. Compared to the previous generation of $C_{soil}$ data HWSD (available only for top 1 m), the current state-of-the-art datasets show significantly higher $C_{soil}$ within the top 1 m (Table 2). On the other hand, the current datasets of vegetation biomass show global $C_{veg}$ ranges from 392 to 437 PgC and substantially lower relative uncertainty than $C_{soil}$. The estimation of the uncertainty that derived from different GPP members shows a range of 100 to 123 PgC (percentile 10 to percentile 90) from different products. Note that the GPP members are different realizations from FLUXCOM and encompass a wide range of sources of uncertainty such as different climate forcing, use of remotely sensed data, and machine learning methods (see Datasets section 2.4). Overall, the results show that the differences in $C_{soil}$ estimates are substantially larger than the differences in $C_{veg}$ and GPP datasets.

Table 2. Estimates of soil organic carbon stocks (Pg C), vegetation biomass (Pg C) and GPP (Pg C yr$^{-1}$).

| Carbon stock in PgC | Non-circumpolar | | | Circumpolar | | | Global | | |
|---|---|---|---|---|---|---|---|---|---|
| $C_{soil}$ | 0-1m | 0-2m | 0-FD | 0-1m | 0-2m | 0-FD | 0-1m | 0-2m | 0-FD |

| | | | | | | | | | |
|---|---|---|---|---|---|---|---|---|---|
| Sanderman | 1218 | 1867 | 2158 | 570 | 996 | 1204 | 1788 | 2863 | 3362 |
| SoilGrids | 1463 | 2404 | 3145 | 925 | 1566 | 1647 | 2388 | 3969 | 4792 |
| LandGIS | 1331 | 2139 | 2731 | 847 | 1570 | 2061 | 2179 | 3710 | 4792 |
| HWSD | 795 | N/A | N/A | 640 | N/A | N/A | 1435 | N/A | N/A |
| NCSCD | N/A | N/A | N/A | 639 | 981 | N/A | 639 | 981 | N/A |
| Mean | 1202 | 2136 | 2678 | 724 | 1278 | 1637 | 1686 | 2881 | 4316 |
| Median | 1275 | 2139 | 2731 | 640 | 1281 | 1647 | 1788 | 3286 | 4792 |
| **$C_{veg}$** | | | | | | | | | |
| Saatchi | 357 | | | 48 | | | 407 | | |
| Avitabile | 368 | | | 35 | | | 404 | | |
| Saatchi-Thurner | 398 | | | 38 | | | 437 | | |
| Santoro | 354 | | | 37 | | | 392 | | |
| Mean | 369 | | | 40 | | | 410 | | |
| Median | 363 | | | 38 | | | 405 | | |
| **GPP** | | | | | | | | | |
| Mean | 104 | | | 6 | | | 110 | | |
| Median | 100 | | | 7 | | | 107 | | |
| P10 | 92 | | | 5 | | | 100 | | |
| P90 | 116 | | | 8 | | | 123 | | |

## 4.2 The spatial distribution of soil carbon stocks

A significant amount of soil organic carbon is stored in high-latitude terrestrial ecosystems, especially in the permafrost region (Hugelius et al., 2013). However, in comparison with low latitudes, the uncertainties of $C_{soil}$ distribution and storage in high latitudes are potentially higher due to fewer available observations of soil profiles. We therefore divided the global soil carbon into the non-circumpolar (Figure 1) and the circumpolar (Figure 2) regions based on the northern permafrost region map of NCSCD. The results show that the mean value and range (maximum - minimum) of $C_{soil}$ in non-circumpolar region (Table 2)

in the top 2m is 2136 PgC and 537 PgC and that in the circumpolar region within the top 2m is 1278 PgC and 574 PgC.

We used in-situ observed soil profiles (Figure S1) and multiple empirical models to select an ensemble of models to extrapolate soil carbon stock to full soil depth (Figure S2 and Table S1). It was apparent that a unique ensemble would be limited to represent $C_{soil}$ profiles globally, resulting in that two different model ensembles were selected to represent the soil vertical distribution, one for the circumpolar regions, and another for non-circumpolar regions. In general, the results show good model

performances for predicting in situ soil carbon stocks up to full soil depth though non-circumpolar regions (Figure S3) show a higher model performance than that in circumpolar regions (Figure S4). The global estimation of $C_{soil}$ to full soil depth results in a higher mean value of 2678 PgC in non-circumpolar region and 1637 PgC in the circumpolar region. Our results show that there are approximately 500 PgC and 400 PgC of carbon stock stored in deep soil layer below 2 meters in non-circumpolar and circumpolar region, respectively.

The spatial distribution of $C_{soil}$ is more consistent across datasets in the non-circumpolar region than in the circumpolar region (Figure 1). The Pearson correlation coefficients (r) between each pair of datasets in the non-circumpolar region are generally higher than in the circumpolar region. Our results show a moderate agreement among the datasets in the spatial distribution of $C_{soil}$ globally (r>0.65). However, there are significant differences in the spatial patterns between the HWSD and each dataset (Figure 1) as the correlation coefficients are all below 0.3. In addition, there is a 2-fold lower carbon storage in the HWSD

than the other datasets. Ratios between the total $C_{soil}$ in the top 100 cm (Figure 1, upper off diagonal plots) show that LandGIS, SoilGrids and Sanderman are consistent in temperate regions but show poor agreement in the tropical and the boreal regions. The comparison also shows that the gradient in carbon stocks between Europe and the lower latitudes diminished in the HWSD soil map. In addition, the spatial distribution and the amount of carbon stocks in insular South East Asia is significantly different in the HWSD.

Higher dissimilarities of spatial patterns across the datasets in the circumpolar region is shown in Figure 2. We included the NCSCD dataset, which specifically focuses on the circumpolar region. The spatial correlations between each pair of the four datasets show low r values, which range from 0.2 to 0.5. In contrast with the non-circumpolar region, the high spatial dissimilarity in circumpolar region indicates higher uncertainty regarding the estimation of total carbon storage. However, there is no evidence on which dataset is more credible in terms of total carbon storage and spatial pattern. The large differences

are possibly due to fewer observational soil profiles in the northern high-latitude regions, which are crucial in the model training process (Hugelius et al., 2013; Hengl et al., 2017).

     The comparison between all datasets shows a good agreement in the vertical structure of terrestrial carbon stocks. The $C_{soil}$ in the top 1-meter is about half of the total terrestrial carbon and 80% for the top 2-meter $C_{soil}$ regardless of region or data source. For the non-circumpolar region, all the datasets show significantly higher carbon storage in the top 1m than that in the HWSD,

while showing less divergence of carbon storage among these three datasets (Table 2). In general, the current datasets show similar vertical distribution of $C_{soil}$ with consistent values and ratios between 1m and 2m soil. The extrapolation results indicate that about 20% of carbon is stored below 2m in the non-circumpolar region. For the circumpolar region, the four datasets show a clear trend that the difference of $C_{soil}$ increases with soil depth, as shown in Table 2. The difference between the top 1m $C_{soil}$ among datasets has a higher difference than that of 2m. However, the ratio between storage in 1m and 2m is similar across all

datasets.

## 4.3 The spatial distribution of vegetation

     Different from the spatial distribution of soil carbon, most vegetation carbon is stored in the tropics whereas much less carbon

resides in the higher latitudes. In fact, the $C_{veg}$ in the circumpolar region is only 10% of that in the non-circumpolar region (Table 2). In comparison with soil carbon, the results show higher consistency and convergence in global estimates of carbon stock among the four global vegetation datasets (Figure 3). Our results show that global vegetation carbon stock is 10% to

25% of the global soil carbon stock, depending on the soil depth considered. The significant spatial correlations (r>0.75, alpha < 0.01) between each of the estimates indicate a consistent global spatial distribution of vegetation across the different data sources. However, the results show more heterogeneity in the regional distribution of vegetation biomass and uncertainty of $C_{veg}$. Specifically, $C_{veg}$ in arid and cold region has higher relative uncertainty than that in the moist and hot regions.

The $C_{veg}$ consists of three components including AGB, BGB and herbaceous biomass. The herbaceous biomass is estimated from mean annual GPP (see Methods 3.2, Carvalhais et al., 2014), and globally represents 5% of the total $C_{veg}$ and less than 1% of the total $C_{soil}$, indicating a minor role of herbaceous biomass in affecting the global estimates and the spatial distribution of $\tau$. The comparison among the four vegetation datasets shows a mean of 410 PgC in $C_{veg}$, with a spread of 11% across the different datasets, and a consistent spatial distribution across the different sources. Locally these differences can be higher, as observed in the relatively higher level of disagreement in sparse vegetated arid and some cold regions (Figure 3, upper off-diagonal subplots).

## 4.4 The spatial distribution of GPP

The global spatial distribution of GPP is similar to that of $C_{veg}$, i.e., high in the tropical regions and low in the higher latitudes (Figure 4). The GPP datasets show high consistency in both the spatial patterns and global values. The spread in GPP estimates is higher (>50%) in arid and polar regions than the other regions (Figure 4, upper off-diagonal plots). Although the differences among different vegetation and GPP estimations, in general, are not as high as in soil carbon, the regionally high uncertainties can be significant.

## 4.5 The ecosystem carbon turnover times and associated uncertainties

The ecosystem turnover time and its uncertainty were estimated using different combinations of $C_{soil}$, $C_{veg}$ and GPP data. We calculated $\tau$ using full soil depth which results in a global estimate of 43 years and ranges from 36 years (25th percentiles) to 50 years (75th percentiles). The uncertainty in the global estimate of $\tau$ is mainly contributed by soil (84%) and GPP (15%) whereas vegetation contributes only marginally (less than 1%). In addition, we derived a global $\tau$ of 37 years and ranges from 31 to 40 years by assuming the maximum active layer thickness to be the full soil depth in the circumpolar regions instead of using only 1-meter $C_{soil}$ as was done in the previous study (Carvalhais et al., 2014). The incorporation of deep soil in the circumpolar region increased the global mean value of $\tau$ by 6 years and uncertainties in the estimations of $\tau$ as well. The global spatial distribution of $\tau$ (Figure 5) shows large heterogeneity, which ranges from 7 years (1th percentile) in the tropics to over 1452 years (99th percentile) in northern high latitudes. The results show a U-shaped distribution of $\tau$ along latitudes where $\tau$ increases nearly three orders of magnitude from low to high latitudes (Figure 7a). Figure 5b shows the map of relative uncertainty that is derived from different datasets. The higher relative uncertainty indicates more spread among the datasets used to estimate $\tau$. Our result shows that $\tau$ estimates at higher latitudes, especially in circumpolar regions, have higher

uncertainties than that at lower latitudes. We found several regions with large spreads in τ among the datasets including north-
east Canada, central Russia and central Australia where the relative uncertainties can span beyond 100%.

### 4.6 The zonal pattern of turnover times

The latitudinal distributions of τ can be best represented by a second-degree polynomial function (Figure 7b). After fitting the
data of all ensemble members, the rate of τ change with latitude can be obtained by taking the first derivative of the fitted
polynomial function. We found that the rate of τ change with latitude has very consistent zonal patterns for different τ ensemble
members from different data sources (Figure 7c). The result shows a consensus on the change of τ with latitude of different
datasets. We also found that the zonal τ gradients were not significantly ($p > 0.05$) different from each other for different
selections of soil depth, indicating soil depth has no significant effect on the τ gradient along latitude. It is worth to note that
there is a significant difference in the zonal τ gradient between the northern and southern hemisphere ($p < 0.0001$) and that τ
increases faster from low to high latitude in northern latitudes than in the southern latitudes. The results show a high confidence
in the zonal distribution of τ and that the difference across datasets does not affect the robustness of the pattern.

### 4.7 The zonal correlation between turnover time and climate

The correlations between τ and temperature and precipitation are analyzed for all the ensemble members at the global scale
(see Methods section 3.5). The τ - T correlation (Figure 8a) is the strongest in northern mid-to-high latitudes between 25º N
and 60º N, and it decreases rapidly from 20º N to the equator. In the southern hemisphere, it increases until 40º S, albeit with
a weaker gradient than in the northern hemisphere. The uncertainties due to differences in ensemble members (shown by the
shaded area) are higher in the transition between the temperate and Arctic regions (50 – 70º N), as well as between tropical
humid and semi-arid regions (20º N to 20º S). Similar to the contribution of different sources to global uncertainty, the spread
in τ - T correlation is mostly due to $C_{soil}$, whereas GPP only affects the zonal correlation to a limited extent (Figure 8c).
However, we find that the τ - T zonal correlation varies negligibly due to data source and soil depth. All ensemble members
agree that τ-T correlation is negative, with stronger associations in cold regions than in warm regions.

The τ - P correlation, in general, has larger variability across latitude and a higher uncertainty related to differences in $C_{soil}$
(Figure 8e). Contrary to the τ - T relationship, the uncertainty of the τ - P correlations from both different data sources and soil
depths are smaller in the tropics than in high latitudes. Negative correlations dominate the high latitudes between 20 and 50º
N and between 20 and 40º S. On the other hand, stronger positive correlations prevail in the tropics. The τ - P correlation
changes the direction from negative in the temperate zone to positive in the tropics, indicating the role of moisture availability
in transitions from arid to humid regions. We also find that the τ - P relationship does not change with different soil depths
(Figure S11).

### 5 Discussion

The accurate estimation of terrestrial carbon storage and turnover time are essential for understanding carbon cycle-climate feedback (Saatchi et al., 2011; Jobbágy et al., 2000). The present analysis benchmarks carbon storage in soil, vegetation and GPP fluxes from multiple state-of-the-art observational based datasets at global scale and provides an estimate of the total carbon stock but also estimates of its vertical distribution and spatial variability. In this section, we will discuss the robustness of the current state-of-the-art estimation on global terrestrial carbon turnover times resulting from the different stock and flux components of τ, and the robustness of its covariation with climate. We first show the variation of spatial and vertical distribution of carbon stock in different regions and the possible reason for the difference, and we then discuss the robustness of zonal distribution of turnover times and zonal changing rates across different datasets. Finally, we focus on the sensitivity of turnover times to climate and potential implications.

## 5.1 Estimation of global soil carbon stocks

We found that there is a significant difference across the current soil carbon datasets in both circumpolar and non-circumpolar regions (Figure 1 and 2). The results show that the uncertainty of $C_{soil}$ estimations in the circumpolar region (52%) is much larger than that of the non-circumpolar region (37%). The spatial patterns of total ecosystem $C_{soil}$ among the soil datasets are more consistent in the non-circumpolar regions, indicating a higher confidence in the current estimation of soil carbon stock in these regions. In contrast with the non-circumpolar regions, there is lower confidence in the circumpolar region in estimating $C_{soil}$ due the fact that there is low spatial correlation across datasets (Figure 1). The difference can be caused by a variety of reasons, e.g.: (i) as an important input to the machine learning methods, in-situ soil profiles are very important factors that influence the final results of the upscaling and using different training datasets can lead to relevant differences in outputs; (ii) the sparse coverage of soil profiles in the circumpolar region may cause the large divergence in the northern circumpolar region. A major difference in the Sanderman soil dataset compared to the other two soil datasets (SoilGrids and LandGIS) is that here the direct target of upscaling was the soil carbon stock, while in the other two datasets the targets were each individual component used to calculate $C_{soil}$ (carbon density, bulk density and percentage of coarse fragments), which were predicted individually. Additional discrepancies may also be associated with the differences in climatic and other input covariates used in the upscaling which may yield a different estimation of $C_{soil}$ (see Section 2.1).

The estimation of a whole ecosystem turnover time is dependent on an estimate of soil carbon stock up to full soil depth. Here, we rely on the available global datasets to follow an ensemble approach for predicting $C_{soil}$ at full depth that selects models with a minimum distance between prediction and observations by using in situ soil profiles (see Supplementary Section S3). The final results depend on the information from the global soil datasets and also on the characteristics of the empirical models. Recent studies have shown the advantage of convolutional neural networks, in comparison to random forest approaches (Hengl et al., 2017; Wheeler et al., 2018), for more robust predictions of soil organic carbon (SOC) with depth (Wadoux et al., 2019; Padarian et al., 2019), which could improve the geographical representation of SOC with depth, although random forests approach already tend to provide unbiased estimates. Overall, the extrapolation provides insights into the carbon storage vertical distribution in deeper soil layers globally, showing that there is approximately 18% of carbon stored below 2

meters globally and over 20% of carbon stored below 2 meters in the circumpolar region. This results from the fact that, in contrast with the non-circumpolar region, the circumpolar $C_{soil}$ does not have a decreasing trend up to 4 meters of soil depth (Figure S1) which indicates that there is a significant amount of carbon stores in deep soil and emphasizes the perspective that deep soil turnover is a key aspect of the global carbon cycle still poorly understood (Todd-Brown et al., 2013).

### 5.2 Consistency in vegetation carbon stocks estimations

Compared with soil carbon, the higher level of consistency in the $C_{veg}$ estimates indicates the stronger agreement on the current estimations in the above-ground carbon components. We show that due to much lower uncertainties in the $C_{veg}$ estimates, the effect of vegetation on the global $\tau$ estimates is minor regardless of which soil depth is used (Table S3). Although the contribution of vegetation to the uncertainties in global $\tau$ estimates is less than 2%, our results show that, locally, vegetation can be the major factor that cause the difference in $\tau$ estimates. As shown in Figure S10, vegetation dominates the uncertainties of $\tau$ in part of the tropics and part of the temperate region in southeast Asia which in total account for 7% of the global land area if only 1m of $C_{soil}$ is used to estimate $\tau$. The land area where $\tau$ uncertainties are dominated by vegetation carbon stocks decreases to 3% and 1%, respectively, when $C_{soil}$ of 2m and full soil depth is considered. Although, our results indicate that vegetation plays a minor role to the global estimates of $\tau$, it is an important factor that can largely affect local patterns of the distribution of $\tau$.

### 5.3 Differences in global GPP fluxes

The contribution of vegetation and GPP to the uncertainties in global $\tau$ is modest compared to the contributions from soil carbon stocks. However, we note that the regional differences in the products can significantly affect the spatial distribution and uncertainty of $\tau$ (Figure 3 and 4). Alternate GPP estimates are likely to impact $\tau$ estimates, although marginally. For example, at global scales, the estimate of a GPP of 123 PgC/yr by Zhang et al. (2017) would lead to a reduction in $\tau$ of ~10% compared to our current estimates (43 years). However, the difference is well within the range of our estimated uncertainty in $\tau$ (~20%) using all the ensemble members. Given the robustness in spatial patterns in GPP estimate from Zhang et al. (2017) compared to the FLUXCOM estimates (r≥0.9, p<0.01, Figure S8), the spatial variability in $\tau$ show a high correlation (r≥0.92, p<0.01) (See Figure S9).

### 5.4 Terrestrial carbon turnover times and associated uncertainties

The current global estimates of $\tau$ are substantially larger than previously (60%), although the global patterns are comparable to previous estimates. Our results show an overall agreement of r = 0.95 between the current estimation and the previous estimation of latitudinal gradient of $\tau$ (Carvalhais et al., 2014). The patterns in the latitudinal correlations between climate and $\tau$ are also qualitatively similar to the previous patterns found, with some particular exceptions in the strength of correlations

between $\tau$ and temperature in northern temperate systems and changes in $\tau$-precipitation correlations, especially in the tropics. A further investigation on the causes behind these differences between the previous and current study reflects that $C_{soil}$ has a substantial contribution to these changes in the correlation between $\tau$ and climate, while GPP has only a modest role in changing the $\tau$-temperature correlation changes in Northern Temperate regions (see Figure S6). This is consistent with the assessment of the largest differences in the spatial distribution of $C_{soil}$ between the three soil datasets used in this study and HWSD soil

dataset used before (Figure 1).

     The uncertainty analysis showed that our current estimation of $\tau$ has a considerable spread which derived from state-of-the-art observations of carbon stocks in soils and vegetation and of carbon fluxes. The uncertainty is mainly stemming from the soil carbon stocks (84%) and GPP fluxes (15%), where the former dominates the vast areas in the circumpolar region and the tropical peatland, while the latter dominates the semi-arid and arid regions (Figure 6). Although GPP shows a strong agreement

in global spatial patterns, local differences between estimates can lead to significant differences in the estimation of $\tau$. This result is consistent with previous observations and model-based studies that also refer to the biases in estimated primary productivity in affecting the carbon turnover estimations to a large extent (Todd-Brown et al., 2013).

     In contrast to global modelling approaches, previous studies have shown that the global soil carbon stocks across observational-based datasets are much less divergent than the ESMs simulations included in CMIP5 (Carvalhais et al., 2014). The CMIP5

results show that the simulated carbon storage ranges from 500 to 3000 PgC, implying a threefold variation in $\tau$ across models (Todd-Brown et al., 2013, Carvalhais et al., 2014). Our current results show that the total amount of carbon in terrestrial ecosystems is substantially higher than the estimation by ESMs, where even the lowest estimation of total carbon storage (in the Sanderman dataset) is about 300 PgC higher than the highest ESM estimation (MPI-ESM-LR, Todd-Brown et al., 2013). The spatial distribution of carbon stocks among ESMs shows a large variation across models (Carvalhais et al., 2014) while

the observational-based datasets are more consistent in the non-circumpolar regions. However, the uncertainty analysis shows that our current estimation of $\tau$ has a considerable spread resulting mainly from the spread in state-of-the-art estimates of soil carbon stocks, followed by the spread in estimates of GPP. The estimation of $\tau$ is dependent on the assumption of a maximum soil depth used to estimate soil C stocks that particularly in the circumpolar regions contributes 54% to the overall uncertainty, while the data source contributes 25%. Soil depth itself is characterized by a large uncertainty given the difficulty in assessing

in-situ measurement uncertainties, in defining a depth at which the soil becomes metabolically inactive, in determining the role of vertical transport to a depth dependent concentration. The challenge in circumpolar regions relates additionally to the influence of active layer dynamics on the spatial and temporal variability in metabolic activity. From an ESM perspective it is difficult to avoid relying on a whole soil, or ecosystem, estimate to compare it with observation-based estimates given that these models abstract from depth dependent soil carbon decomposition dynamics, or have not reported depth of the soil carbon

stocks (Carvalhais et al., 2014). In this aspect, an explicit consideration of soil C stocks at depth in ESMs would be instrumental in understanding and evaluating the distribution of ecosystem carbon stocks and turnover times against observations.

     It is worth noting that here the estimation of $\tau$ is based on the steady-state assumption, that is, the assumption of a balance net exchange of carbon between terrestrial ecosystems and the atmosphere. Here, the assumption is that integrating at larger spatial

scales, by averaging the local variations in sink and source conditions, reduces the differences between assimilation and out-

fluxes relative to the gross influx; and that the integration of stocks and fluxes for long time spans reduces the effects of transient changes in climate and of inter-annual variability in τ estimates. However, this assumption is valid to a much less extent at smaller spatial scales (site-level) and shorter time intervals, as the ecosystem-atmosphere exchange of carbon is most of the time not in balance and forced steady state assumptions can lead to biases in estimates of turnover times and other ecosystem parameters (Ge et al., 2018; Carvalhais et al, 2008).


## 5.5 Robust associations of τ and climate

Despite the large uncertainty in the τ estimations, we identified robust patterns on the τ-climate relationship that can be instrumental in addressing the large uncertainties in modelling the sensitivity of terrestrial carbon to climate, which are reflected in the spread of τ estimates by the different ESMs (Tod-Brown et al., 2013). The zonal distribution of τ is a robust

feature that changes little across different datasets, which indicates that the current state-of-the-art datasets all agree on the latitudinal gradient of the carbon turnover time (Figure 7). In addition, the latitudinal change rate of τ is robust against any considered soil depth (Figure 7), which reflects pattern comparability between assumptions of τ gradients up to one meter (Koven et al. 2017; Wang et al., 2018) or to full soil depth (Carvalhais et al., 2014). The robustness on the latitudinal patterns in the ensemble are likely to emerge from the latitudinal gradient in temperature, shaping the zonal distribution of τ that

increases towards the poles as mean annual temperatures substantially decrease.

This study addresses the robustness in the τ-climate association by investigating the zonal correlations between τ and temperature and between τ and precipitation. The τ-temperature correlation varies with latitude where high correlations are found at higher latitudes and low to moderate correlations found closer to the tropics (Figure 8). The latitudinal gradient in the τ - T relationship is similar when compared with previous results (Carvalhais et al., 2014) although the strength of the

correlations can vary marginally by changing GPP products, but more substantially when exchanging the $C_{soil}$ datasets (Figure S6). However, these relationships show strong robustness across state-of-the-art datasets (Figure 8). On the other hand, the zonal patterns of τ-precipitation are more challenging to converge across different $C_{soil}$ sources (Figure 8e) when compared with uncertainties stemming from GPP (Figure 8f) regardless of depth considered (Figure S11). Overall, the correlation between turnover times and precipitation in the tropics is higher than that with temperature as shown in Figure 8d, indicating

a potentially more dominant role of precipitation in the tropics (Wang et al., 2018).

Overall, the τ-P correlations, although varying in strength, are robust across the data ensemble except when controlling for $C_{soil}$ source (Figure 8e). The role of $C_{soil}$ in the τ-P relationships is independent of depth (Figure S11) and explains most of the differences found in the patterns to previous results (Carvalhais et al., 2014), which are mainly caused by the differences in the soil carbon stock (Figure S6). Given that the data and methodological support are substantially shared across the different

approaches (see Data section 2.1) and potential limitations in representing contributions of soil moisture to τ at deeper layers, even shallower than 2m, these results highlight the relevance of better understanding and diagnosing the effects of the

hydrological cycle on $\tau$. The limitation may be linked to the realization that random-forests-based methods tend to show high correlations between predicted top soil and deeper soil estimates of $C_{soil}$, and also lower correlations to deeper $C_{soil}$ geographic variability (Wadoux et al., 2019; Padarian et al., 2019).

Ultimately, given the recognition that the sensitivity of the terrestrial carbon to climate is a major uncertainty reflected in the spread of $\tau$ across different ESMs, the reliable estimation of $\tau$ and identification of robust patterns in $\tau$-climate associations is key to provide robust constraints to improve the performance of the current ESMs. Notwithstanding, the intimate interaction of energy and water along with other factors such as land use change all affect $\tau$ but on different spatial and temporal scales. Further research directions would gain by exploring the contribution of addition potential factors that may influence the spatial
distribution of $\tau$, such as mortality and disturbance regimes, human impact via management regimes or land cover change dynamics, and the vertical distribution of the hydrological cycles.

## 6 Data availability

The dataset of whole ecosystem turnover times of carbon presented in this study can be downloaded from the Data Portal of
the Max Planck Institute for Biogeochemistry at https://doi.org/10.17871/bgitau.201911 (DOI: 10.17871/bgitau.201911).

## 7 Conclusion

A full assessment of the global turnover times of carbon is provided using an observational-based ensemble of current state-
of-the-art datasets of soil carbon stocks, vegetation biomass and GPP. At the global scale, the uncertainties in $\tau$ estimates are dominated by the large uncertainties in soil carbon stocks. The uncertainty of carbon stocks and $\tau$ estimation in the circumpolar region is significantly higher than that in the non-circumpolar region. Our results show that there is a consistent vertical distribution of soil carbon across datasets, and it is estimated that soils below 2 meters take up to 20% of total soil carbon globally. A spatial analysis shows that both soil carbon and GPP are the major contributors of local uncertainties in $\tau$ estimation.
The differences in soil stocks between datasets dominate the uncertainties of $\tau$ in the circumpolar region, while the spread in GPP dominates the uncertainty in semi-arid and arid regions. The difference in vegetation data has a minor contribution to the uncertainty.

Despite the differences, we identified several robust patterns that change only marginally across different ensemble members of $\tau$ that derived from different datasets or different soil depths. First, we found a consistent latitudinal pattern in $\tau$ that can be
described by a second-degree polynomial function. The changing rate of $\tau$ with latitude can be described equally well for all ensemble members and the changing rate of $\tau$ with latitude is highly consistent across different datasets and does not change with soil depth. The same zonal correlations between $\tau$ and climate showed there is a robust association of $\tau$ with temperature and with precipitation. However, we note that association between temperature/precipitation and $\tau$ change with latitude.

Specifically, temperature mainly affects the τ variation in middle to high latitudes beyond 20ºN and 20ºS while precipitation affects τ not only in temperate zones but also in the tropical regions. Overall, this study synthesizes the current state-of-the-art data on global carbon turnover estimation and argues that the zonal distribution of τ and its covariation with climate is robust across the diverse observation-based ensemble considered here. These results build on previous effort and support exercises for benchmarking ESMs.

## Author contributions

NF and NC designed the study. NF conducted analysis and wrote the manuscript under the orientation of NC. MT, VA, and MS provided data for the analysis. NF and UW collected and harmonized datasets. NF, UW and SK contributed to methodological development. SK participated in the discussion and development of the paper. All authors contributed to the discussions and interpretation of the results and the writing of the manuscript.

## Competing interests

The authors declare no conflict of interest.

## Acknowledgments

We would like to acknowledge the support of Tomislav Hengl for providing soil carbon data and discussions regarding the analysis. We would like to thank Saatchi Sassan for providing one of the vegetation biomass datasets. We thank Martin Jung for providing GPP data and useful feedback; and Jacob A. Nelson for suggestions on improving the manuscript language.

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

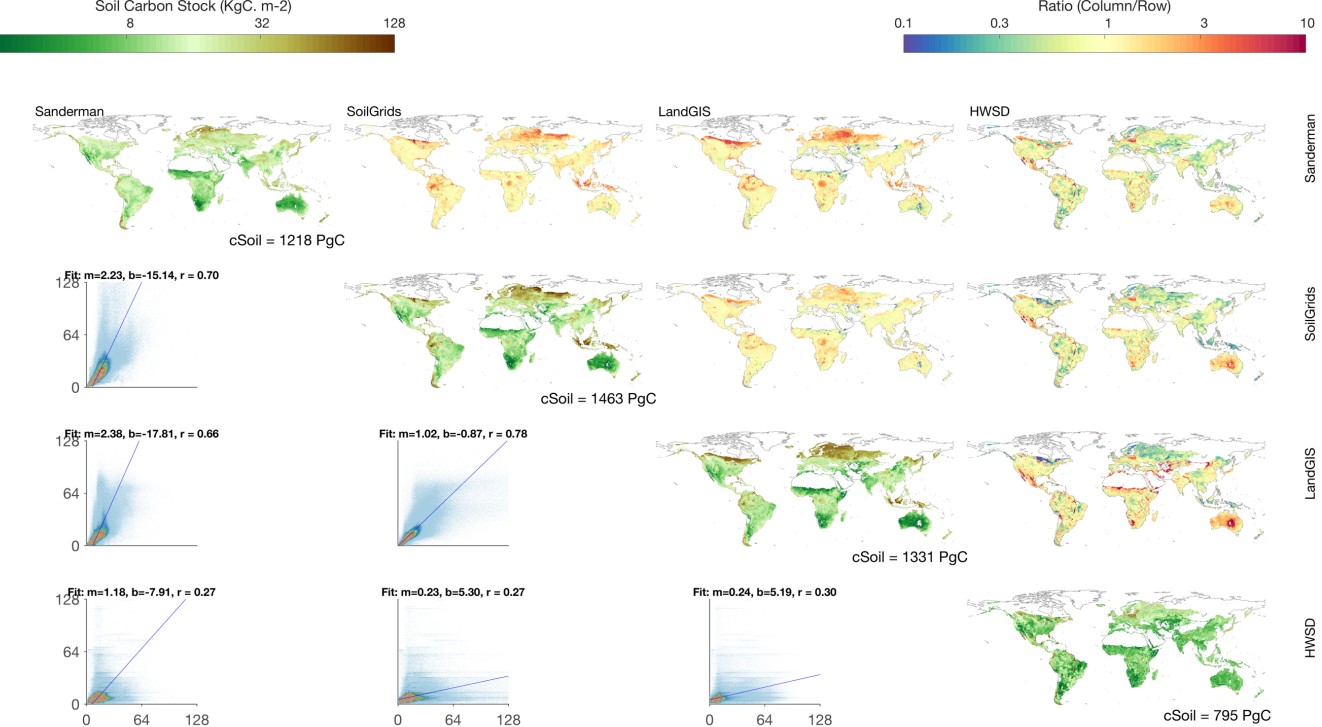

**Figure 1: Spatial distributions of soil carbon storage at 0-100cm in the non-circumpolar region.** The total amount of carbon stock is shown in the bottom of each diagonal subplot. The upper off-diagonal subplots show the ratios between each pair of datasets (column/row). The bottom off-diagonal subplots show the density plots and major axis regression line between each pair of datasets (m: slope, b: intercept, r: correlation coefficient). The ranges of both of the colorbars approximately span between the 1st and the 99th percentiles of the data. Hereafter, all figures comparing different spatial maps include the information in a similar manner.


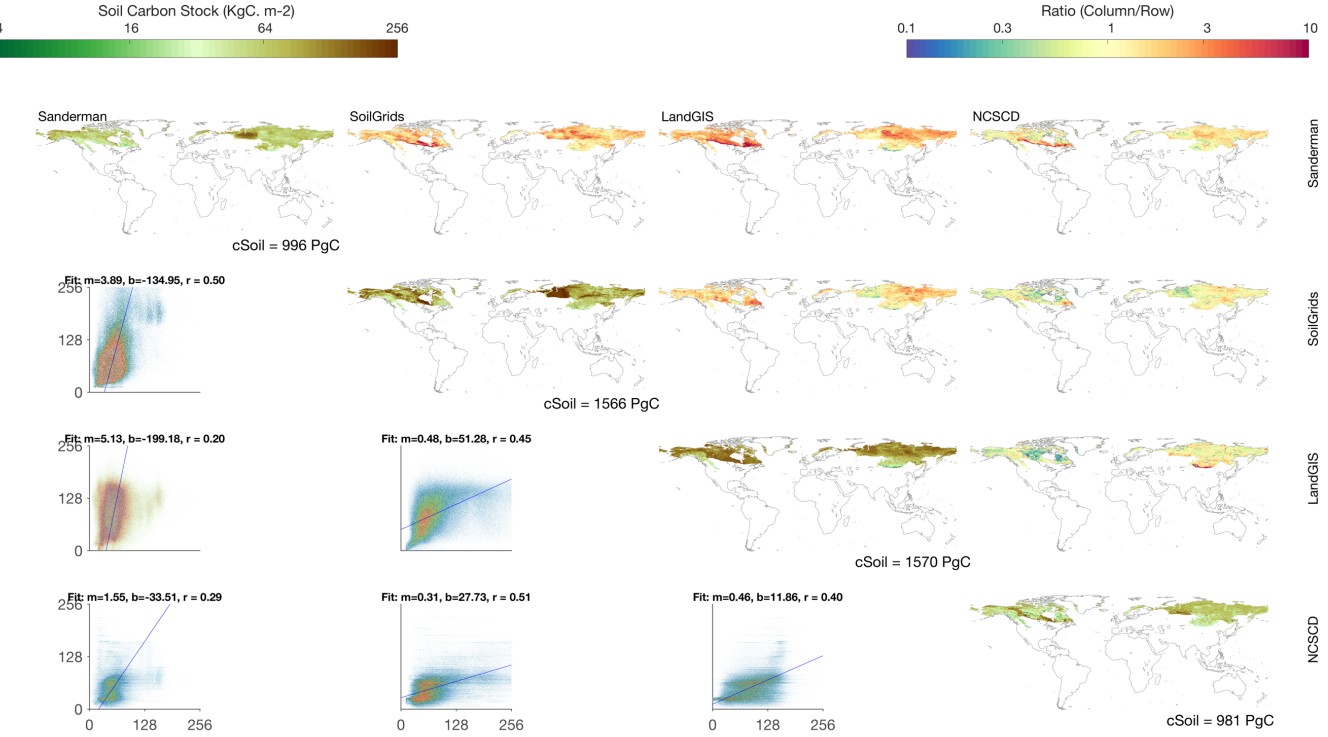

 **Figure 2: The same as Figure 1 except for the C<sub>soil</sub> in 0-200cm and in the circumpolar region.**



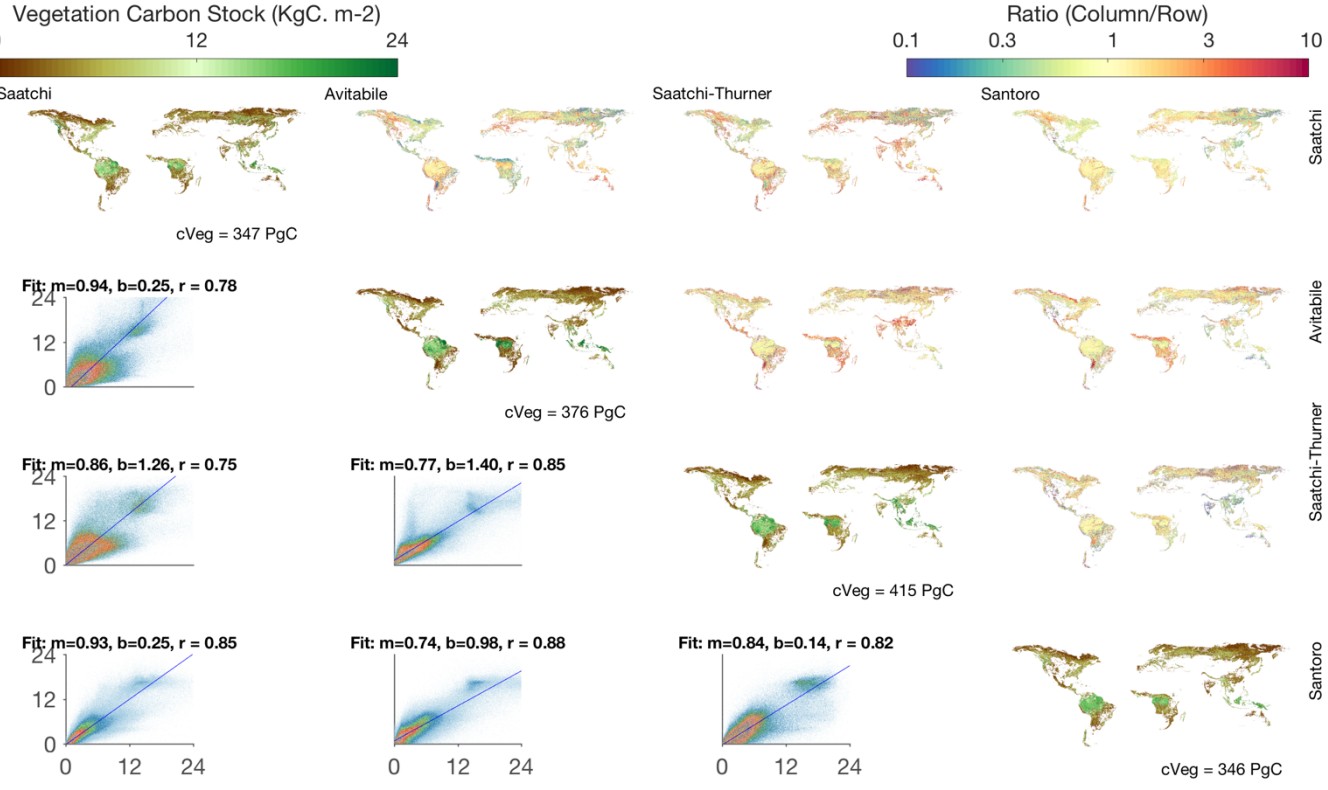

**Figure 3: The same as Figure 1 but for vegetation carbon stocks.** The total vegetation carbon stocks is calculated as the sum of aboveground (AGB), belowground (BGB), and herbaceous biomass. For consistency, only the grid cells where all four maps have values are included. Therefore, the total amounts in the diagonal subplots differ slightly from those in Table 2.

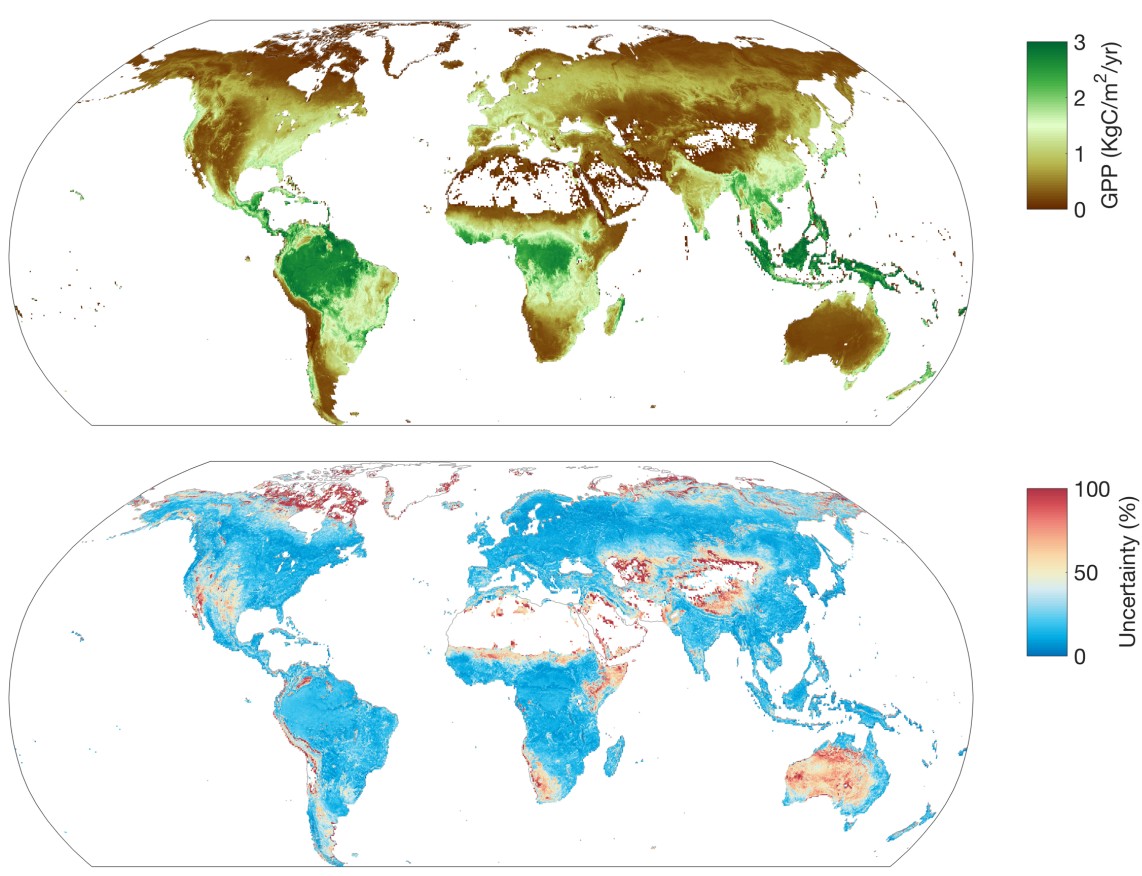

**Figure 4: Spatial distributions of GPP and its uncertainty.** The upper panel shows the spatial distribution of mean annual GPP, the lower panel shows the relative uncertainties (calculated as a ratio of interquartile range to mean). The ranges of both the colorbars approximately span between the 1st and the 99th percentiles of the data.

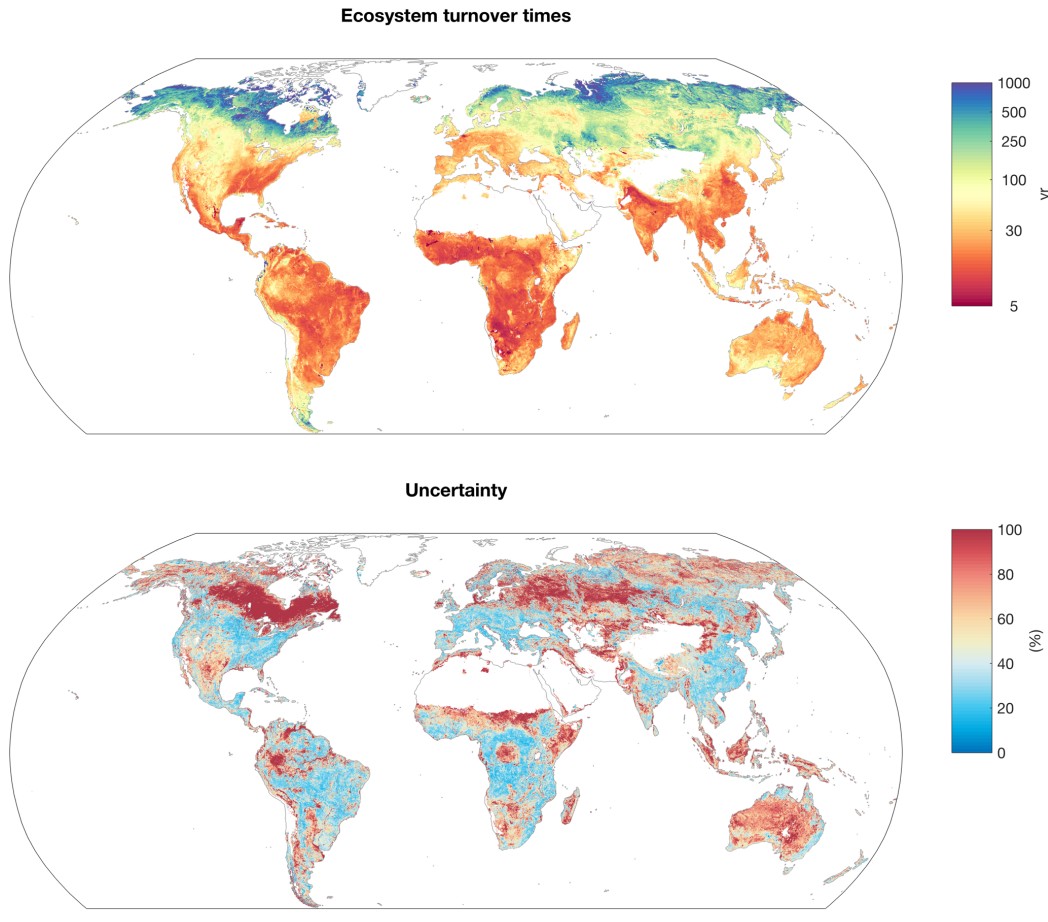

**Figure 5: Spatial distribution of ecosystem turnover times.** The upper panel shows spatial distribution of turnover times, and the lower panel shows the relative uncertainty (calculated as a ratio of interquartile range to /mean). The range of colorbar in the upper panel approximately spans between the 1st and the 99th percentiles of the data, and the one is lower panel spans between the 1st and 90th percentiles.

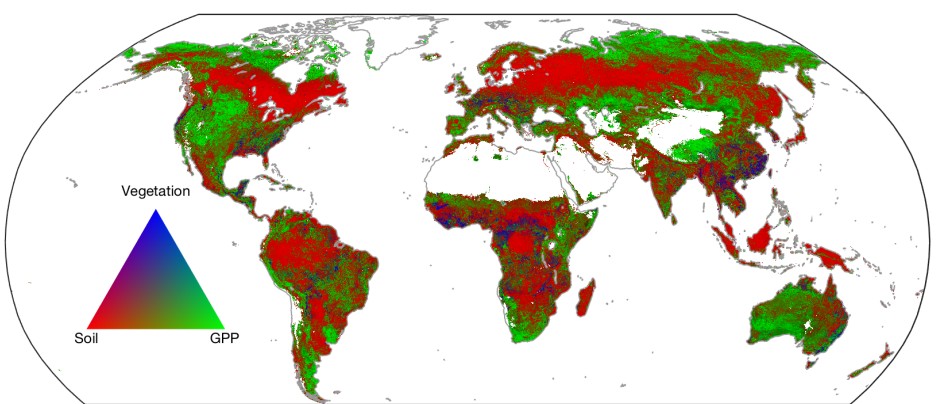

**Figure 6: The sources of τ uncertainty.** The contribution of different sources of soil (at full soil depth), vegetation and GPP data to the uncertainty in turnover time. The green color indicates the regions where the uncertainty is dominated by GPP, red by soil carbon, and blue by vegetation carbon. Soil, vegetation and GPP dominates 64.8%, 32.4% and 2.7% of land area in the uncertainty of τ.

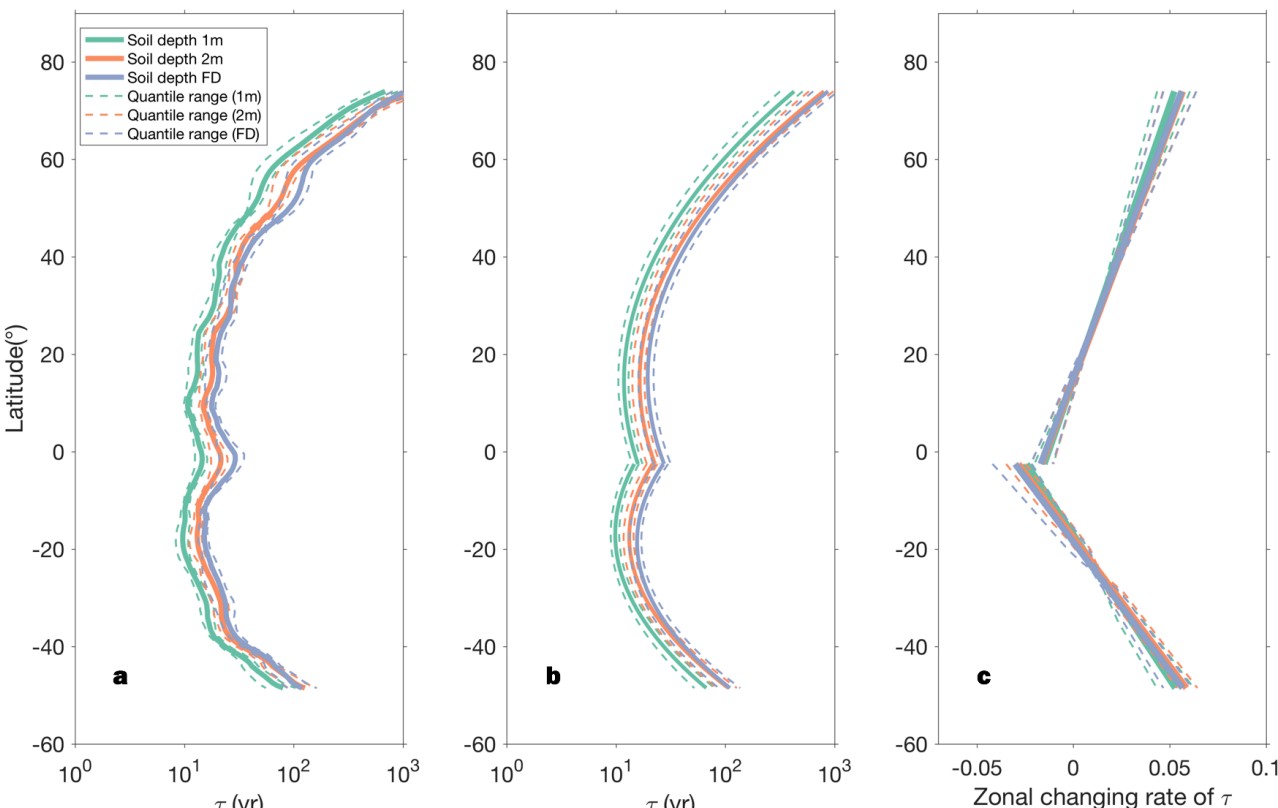

Figure 7: (a) The zonal distribution of τ. (b) Second-degree polynomial fit to the zonal distribution of τ. (c) Zonal rate of changes of τ with latitude (calculated as the first derivative of the polynomial function). Solid lines represent the mean τ for different soil depths (1 m, green; 2 m, red; full depth, purple) and dashed lines in corresponding colors are the interquartile range. The polynomial function is fitted independently for the northern and southern hemispheres. The latitude that divides the northern and southern hemisphere is located at 2°S where there is a local maximum of zonal τ in a).

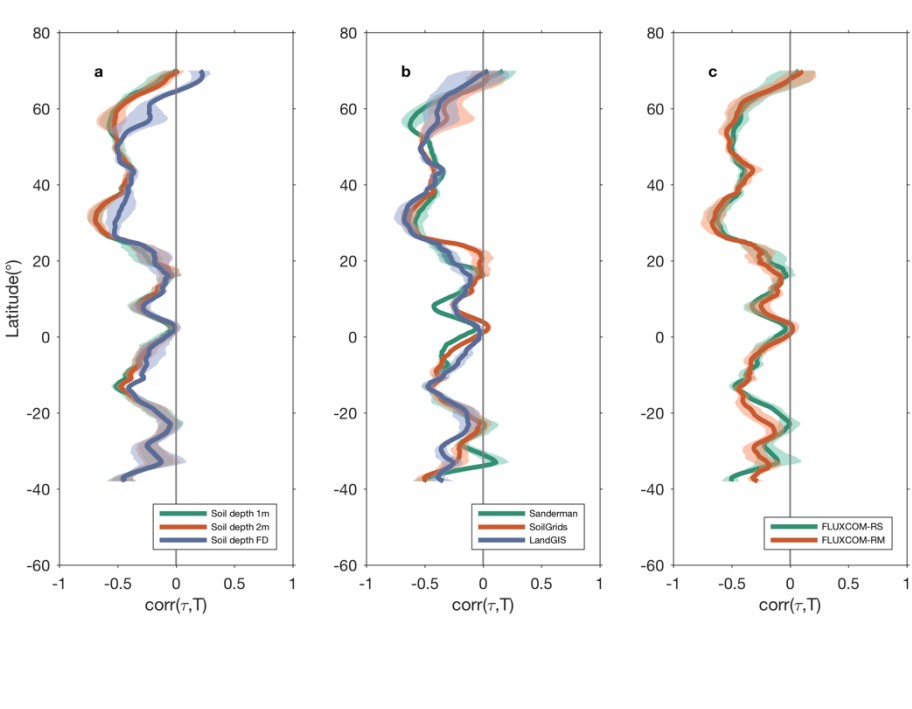

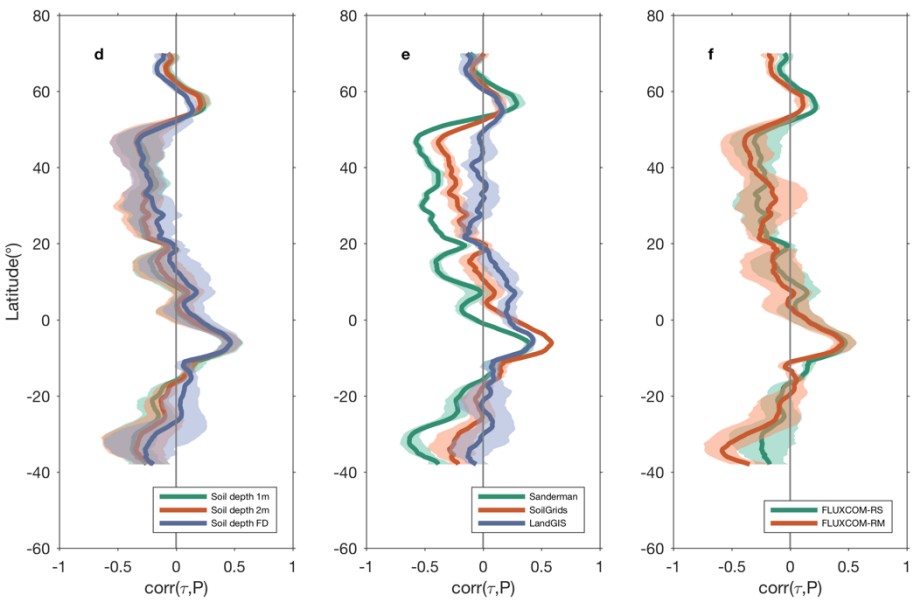


**Figure 8: Correlation between zonal $\tau$ and mean annual temperature (T)/mean annual precipitation (P).** Subplots (a) and (d) are colored by different soil depth (1m, green; 2m, red; full soil depth, blue) with shaded areas of interquartile range. Subplots (b) and (e) are colored by different soil sources; Subplots (c) and (f) are colored by different GPP products of different forcing (remoting-sensing only and remote-sensing + meteorology). The correlations are consistent across the different latitudinal span widths considered (see Methods Section 3.5) and hence not shown here.
