# Peer review of "Apparent ecosystem carbon turnover time: uncertainties and robust"

_Earth System Science Data, 2019_

## Referee Comment (RC1) · Anonymous Referee #1 · 11 Mar 2020

Review of "Apparent ecosystem carbon turnover time: uncertainties and robust features"

General comments

Fan et al., have estimated the bulk ecosystem level (i.e. combined vegetation and dead organic matter) carbon turnover (or residence) time at global scale. Fan et al., combine multiple remotely sensed estimates of above ground woody biomass with observation orientated estimates of gross primary productivity (GPP) and soil carbon stocks at different depths. Additional information is estimates from these data to estimate the total woody carbon content (though estimating the below ground) and carbon content of herbaceous layer and soil carbon at its maximum depth. The analysis provides and update to a previous study led by co-author Carvalhais from 2014 with the primary

update appearing to be the inclusion of the impact of soil carbon stocks at greater depths than those typically considered in soil inventory or simulation model (i.e. >2 m depth).

The paper provides a novel combination of datasets and a high quality spatially explicit estimate of uncertainty for their estimate which provides valuable information. However, that is not to say the manuscript is without issues.

I would be interested to hear more information about some of the derived datasets. For example, the creation of the herbaceous carbon stock map is described but what is the relative proportion of vegetation carbon found within the herbaceous layer is not stated? As the GPP ensemble is used in the estimation of the herbaceous layer what is the uncertainty in the herbaceous carbon content? How does the herbaceous carbon stock influence ecosystem turnover time vary in space, i.e. could it have been neglected? Similarly, the soil carbon estimated to maximum depth would be interesting to investigate further. A really simple but nice addition would be a map of the maximum soil depths inferred by your analysis.

The current text is a little unbalanced towards Csoil sometimes to the exclusion of Cveg or GPP in the introduction, results and discussion sections.

The introduction sets out the overall challenge and usefulness of such datasets in constraining Earth System Models and their role in quantifying the response of the terrestrial ecosystem to climate change. However, the fact that this is an update paper is not made fully clear. Doing so would I think make it straight forward to highlight the weaknesses of the previous analysis and how they are being improved here making a more robust and unique dataset. I honestly do support making updates and improvement to existing datasets as this provides a clear traceable advancement in the science. Because the current manuscript does not clearly highlight soil as a weakness / uncertainty of existing works the introduction reads as being very soil dominated with little introduction of the vegetation carbon stock challenges or the estimation of GPP.

The introduction does clearly state one of the key assumptions, that ecosystems are assumed to be in steady state. What is missing is an appreciation that much of the worlds vegetation is not in steady state, either due to direct human intervention (biomass removal or other land use change) or as a result of increasing $CO_2$ concentration. Attempting to quantify this is out of scope but I think it would be useful to include either in the introduction or discussion the potential implications of this assumption leading to an underestimate in turnover times (e.g. Ge et al., 2018).

The methods are thorough and cover each of the input data products and associated methods.

The results section, like the introduction, seems to be biased towards soil carbon results rather than a complete overview. This should be addressed. Further information can be found below in the technical comments.

The discussion lacks any discussion of the vegetation carbon stocks and almost any discussion of the GPP estimates. I also find it odd that figures 1-4 are not mentioned in the discussion at all. The discussion lacks sufficient comparison with existing studies / ESM outputs which this dataset should be constraining. One exception being the comparison with Todd-Brown et al., 2013 comparing soil carbon turnover times from CMIP5 models. Discussion of GPP importance is limited to its uncertainty contribution in the current analysis. While I have no problem with your choice to use FLUXCOM GPP estimates as observation-orientated. I do think it would be useful to include some discussion / context that compares your GPP estimate to alternate approaches e.g. remote sensing products (e.g. Zhang et al., 2017) or terrestrial ecosystem models constrained with remote sensing (e.g. Norton et al., 2019).

Overall, this paper should be published after some improvements to the manuscript text. The analysis is novel in that it introduces new information to update an existing assessment. The study is also rigorous through the use of multiple datasets and quantification of their collective and partitioned uncertainties on ecosystem turnover time.

[Figure]

The estimate of global turnover and time its uncertainty is also useful, providing constraint on the bulk ecosystem turnover against which other approaches and models can be compared.

Technical comments

Abstract

L14: "...controls..." -> "...is an important determinant of..." Turnover time is not a singular control.

L14-15: "...poorly simulated..." as this paper itself shows there is still plenty of uncertainty in turnover time estimate not just ESMs please rephrase.

L16: "...new, updated ensemble..." Somehow this reads slightly odd to me. I am not sure you should say both new and updated. I think it is clearer to say that you have created a state-of-the-art update to an existing map.

L19: what confidence level are the uncertainties given at? Same for L21.

L19: "...longer than the previous..." at the moment it has not been made clear what the previous is.

L22: remove "merely"

L22: "Cveg (0.05 %)" I find this very surprising and I think others will too. You need to support this somehow, e.g. showing the relative difference in the uncertainty of Cveg estimates vs Csoil. Also, the uncertainty proportions reported leave ~20 % unaccounted for. This should be made clear and some hypotheses as to what might account for this is useful.

L24: "...full depth Csoil..." at the moment it is not clear what this means. As in full depth relative to what? Obviously in the context of the overall paper this is compared to assuming soil depth of 1 or 2 m. Somehow this needs to be made clearer in the abstract.

[Figure]

L29-32: "Our findings show that the..." consider moving these statements further up in the results component of the abstract as I think this is the take-home information. So I would make a bigger deal out of it.

Introduction

L37: "Terrestrial ECOSYSTEM carbon turnover time"

L39: "Ecosystem turnover time is an emergent..." I would suggest that it is a good idea to quickly reinforce the research object to the reader.

L39: "...better..." better than what? Should be made clear.

L41:43: Some context on the steady state assumption needed either here or in the discussion.

L49:55: Introduces the importance of ecosystem turnover and its climate sensitivity to the response to climate change. But only soil carbon stocks mentioned. There should be some introduction of each of the main components just mentioned in the previous paragraph, i.e. C update via photosynthesis, Cveg and Csoil. Friend et al., 2014 (cited in text) does cover vegetation simulation in models so you may not even need a new reference, just fill out the text.

L81: "global estimate of ecosystem turnover time" at what spatial resolution?

Datasets

L100: "availability of field data"

L108: "The dataset..." not clear which dataset. SoilGrids, S2017 or both?

L112: "PH"-> "pH"

L167: "Ge et al., 2014" not in reference list

L175-180: How many ensemble members in the FLUXCOM experiment? I think it would be good to give information on the ensemble mean uncertainty in absolute and

relative terms. The final statement ". . .we derived the long-term mean. . ." also makes it slightly ambiguous as to whether you also averaged across the ensemble. Given you have quantified the uncertainty I know that is not the case, but I would revise the text here to make that clear.

Methods L207: herbaceous carbon stock estimation, is the uncertainty in GPP propagated here too?

L214: Similarly, to me the manuscript is not clear whether the statistical uncertainty is propagated into the Cveg estimates?

L220: ". . .was used to optimize parameters of the models." A reference is needed for the approach or the software and package used to do this.

L223-234: I am not clear from this description whether the extrapolation process was estimating the cumulative C stocks down to a predetermined maximum soil depth from a database or whether maximum soil depth emerges from the analysis?

Results

L255: I would clarify to the total number of ensemble members of ecosystem turnover time which has been created.

L263: ". . .and has a SMALLER relative uncertainty THAN. . ." I would be explicit that Cveg uncertainty at global scales is smaller than Csoil

L264: Be clear here and remind the reader that the different GPP products / estimates are all from FLUXCOM.

L266: Table 2. I would like to see the herb fraction or total given here along-side the Cveg.

L272: I suggest you include some typical uncertainty values of Csoil at high and low latitude to give context in the text.

Consider whether Section 4.2 and 4.3 should be merged or re-arranged (and titled) to make what they are actually discussion clear. As it is both "regional" and "spatial" headings suggest similar things.

Sect 4.4. It titled "global carbon stocks" but includes only soil.

There is no similar paragraph presenting the results of the other components of the analysis. There may not be much interesting to say about them but at the moment it looks odd to focus on soil without explanation to the lack of results on other components.

L310: Given the explicit comparison made here to the original 2014 paper. A clear and direct spatial comparison be of the previous map and the current may be useful.

L312: "higher"->"longer" time cannot be higher.

L328-329: "For instance, the uncertainty contribution from Cveg becomes smaller…" Does the spatial pattern in relative contribution between Cveg and Csoil persist despite the change in magnitude?

L331-332: "Overall,…" seems like the headlined result for the paragraph, should it not have come first with the details coming afterwards. Also these numbers appear to be different from those quoted in the abstract. Could you clarify?

L349: "Figure 6a and 6d)" There is no figure 7.

Discussion

L369: "…understanding carbon cycling-climate feedbacks (REF)"

L368-370: I would be clear over how many products you have which are to be made available. As I suggested earlier, that providing the derived datasets could be useful.

Sect. 5.1 titles for "global carbon stock" The entire section deals with Csoil alone. No Cveg.

L375: "...non-circumpolar region (Figure X)"

L384-385: This appears to be new information introduced in the discussion. You should introduce all your results in the results section first.

L386: "...global carbon cycle yet poorly understood (REF)."

L388: "Two model ensembles..." be careful with what you are referring to as an ensemble. There are multiple in the manuscript, the turnover time itself, GPP estimate.

L388-389: "Two model ensembles were selected that can best represent the soil vertical; distribution in circumpolar and non-circumpolar regions RELATIVE TO IN-SITU OBSERVATION".

L396-403: A comparison with ESMs is good to have here. But As the models only simulated to 1 or 2 m depth. I think it would be fair to compare how the models agree with the soil C stock to that depth too. The question over to what soil depth we should consider needs to be discussed too. For example, at what depth does the soil become metabolically inactive? In high latitudes soil carbon does not turnover once it is frozen so a couple sentences highlighting the importance of the active layer depth would be interesting context. I know this is mentioned in one sentence in the next section but there is no numbers given or reference.

L405-409: Somewhere in here a couple lines to discuss the potential importance of different GPP estimates which are often much higher than those estimates by FLUX-COM would be appropriate. Again, I do not think that this undermines your analysis as FLUXCOM is an observationally orientated estimate but the context that other estimates can provide much larger GPP values. For example, you can highlight how the tendency for larger GPP estimates in ESMs will lead to errors in the turnover time estimation.

L410: "..remains inactive in the process of turnover (REF)" Reference needed and expand.

L418-420: Might be useful to indicate the typical range of soil depths simulated to by the current generation of models in CMIP6.

L424-425: Rephrase

L427: "...to quantify ITS CLIMATE sensitivity" be specific to improve clarity.

References

Ge et al., (2018) https://doi.org/10.1111/gcb.14547

Norton et al., (2019) GPP = 167 +/-5 PgC/yr https://doi.org/10.5194/bg-16-3069-2019, 2019

Zhang et al., (2017) GPP = 121.60 to 129.42 Pg C/yr https://doi.org/10.1038/sdata.2017.165

---

## Referee Comment (RC2) · Anonymous Referee #2 · 11 Mar 2020

This paper describes a significant undertaking in a critical ecosystem property, i.e. terrestrial carbon turnover time. The dataset production process and the relevant points are described clearly in the paper. The dataset will be very interesting and useful to ecological modelers, although I did not have the chance to review the dataset because I could not download the dataset somehow. I recommend publishing the article in ESSD after addressing the minor issues listed below.

General comments: 1) The dataset can only be downloaded when the users registered on the website. After I registered, somehow I still cannot download the dataset. So, I only reviewed the manuscript not the dataset. Whether the original data and the process data used to derive the turnover time can also be downloaded from the link? This would be helpful for people trying to reproduce the data generation process or

[Figure]

for those that would like to use original data or process data. 2) The turnover time was estimated assuming steady state, in which the efflux equals to the influx. While the reality is in non-steady state. The effects of this assumption on the estimation of turnover time should be discussed. 3) Was the high consistency of vertical structure of soil carbon storage caused by the consistent extrapolation model? i.e. same model parameters lead to the same vertical ratio? (P15L393) 4) How to compare the sensitivities of turnover times to precipitation and temperature? They have different units (P16L430). 5) The influence of other factors on turnover times are missing. Could you give further results or discussion? (P16L435) 6) The GPP only used one data source, i.e. FLUXCOM produced by Jung. There are also other sources of GPP such as the GPP generated using LUE model published in Nature Scientific Data. It would be interesting to see the change in uncertainty.

Specific comments: P6L188: The R and r is not consistent. P10L256: The vegetation biomass is missing in the first sentence. P14L378: "caused" should be "caused by". P16L436: Why the relationship between turnover time and precipitation are different with previous studies? P16L447: Typo. Should be "state-of-the-art". P19L570: The color of this reference is different from other parts of the manuscript. Fig. 1 and Fig. 2: It should be noted that the bottom diagonal subplot was the regression of row with column, i.e. y=row, x=column? Besides, what did the color around the origin represent? Fig. 3: Quantile range here is 25Fig. 5: How to determine the turning point? It seems like not 0? Fig. 6: The lines in subplot c and f indicate? Terminology: The soil dataset provided by Sanderman et al 2017 was noted as S2017 in the text and the tables, while in the figures it was noted as Sanderman. Please be consistent through the manuscript. supplement-P2L32: CO2 should be $CO_2$. supplement-P3L59: The period was missing between "Table 2" and "All".

---

## Author Comment (AC1) · 31 May 2020

Dear referee,

Thank you for making these constructive comments on our manuscript! I am glad you like the results we want to present. The responses to your comments can be found below. Please also take note on the marked-up manuscript in the attachment, where all the changes from the original submission are highlighted.

The responses:

I would be interested to hear more information about some of the derived datasets. For example, the creation of the herbaceous carbon stock map is described but what is the relative proportion of vegetation carbon found within the herbaceous layer is

not stated? As the GPP ensemble is used in the estimation of the herbaceous layer what is the uncertainty in the herbaceous carbon content? How does the herbaceous carbon stock influence ecosystem turnover time vary in space, i.e. could it have been neglected? Answer: We added some analysis of herbaceous biomass in the text per request. Please see the updated manuscript in detail but the simple answer is that the herbaceous biomass plays a minor role in the estimation of $\tau$ since it is less than 1% of soil carbon stock and 5% of vegetation carbon stock.

Similarly, the soil carbon estimated to maximum depth would be interesting to investigate further. A really simple but nice addition would be a map of the maximum soil depths inferred by your analysis. Answer: We added the soil depth global distribution map (Figure S6). Please also note that the full soil depth is not inferred by our analysis but a global dataset (see Method). The current text is a little unbalanced towards Csoil sometimes to the exclusion of Cveg or GPP in the introduction, results and discussion sections. The introduction sets out the overall challenge and usefulness of such datasets in constraining Earth System Models and their role in quantifying the response of the ter- restrial ecosystem to climate change. However, the fact that this is an update paper is not made fully clear. Doing so would I think make it straight forward to highlight the weaknesses of the previous analysis and how they are being improved here making a more robust and unique dataset. I honestly do support making updates and im- provement to existing datasets as this provides a clear traceable advancement in the science. Because the current manuscript does not clearly highlight soil as a weakness / uncertainty of existing works the introduction reads as being very soil dominated with little introduction of the vegetation carbon stock challenges or the estimation of GPP.

Answer: Yes, we updated the manuscript to a more balanced way. Please see details in the line to line answer below and the marked-up manuscript.

The introduction does clearly state one of the key assumptions, that ecosystems are assumed to be in steady state. What is missing is an appreciation that much of

the worlds vegetation is not in steady state, either due to direct human intervention (biomass removal or other land use change) or as a result of increasing CO2 concentration. Attempting to quantify this is out of scope but I think it would be useful to include either in the introduction or discussion the potential implications of this assumption leading to an underestimate in turnover times (e.g. Ge et al., 2018). Answer: Yes, we updated the manuscript to address this matter. Please see details in the line to line answer below and the marked-up manuscript.

The results section, like the introduction, seems to be biased towards soil carbon results rather than a complete overview. This should be addressed. Further information can be found below in the technical comments. The discussion lacks any discussion of the vegetation carbon stocks and almost any discussion of the GPP estimates. I also find it odd that figures 1-4 are not mentioned in the discussion at all. The discussion lacks sufficient comparison with existing studies / ESM outputs which this dataset should be constraining. One exception being the comparison with Todd-Brown et al., 2013 comparing soil carbon turnover times from CMIP5 models. Discussion of GPP importance is limited to its uncertainty contribution in the current analysis. While I have no problem with your choice to use FLUXCOM GPP estimates as observation-orientated. I do think it would be useful to include some discussion / context that compares your GPP estimate to alternate approaches e.g. remote sensing products (e.g. Zhang et al., 2017) or terrestrial ecosystem models constrained with remote sensing (e.g. Norton et al., 2019). Answer: Yes, we updated the manuscript to a more balanced way. Please see details in the line to line answer below and the marked-up manuscript.

Response to the technical comments: Please check the modified manuscript if the answer after the number of lines is 'revised'. Otherwise, the specific modifications or reasons are listed. L14: revised. L14-15: revised. L16: revised. L19: the interquartile range. L19: added the reference. L22: revised. L22: Actually, the supporting results are shown in section 4.5 and Figure 4. Yes, this is probably a little bit surprising that Cveg doesn't contribute much to the global uncertainty. This is because the uncertainty

in soil carbon stock is dominant. L24: Added reference. L29-32: Great suggestions! The regarding part is moved up. L37: revised. L39: Indeed, revised. L41-43: They are added in the discussion. Please see Section 5.2. L49-55: revised. L81: revised. L100: revised. L108: revised. L112: 'PH' is deleted because of not related to the context. L167: revised. L175-180: Yes, it is added in Section 3.1. L214: No, but we propagated the uncertainty of different vegetation datasets into the turnover estimations. L220: revised. L223-234: revised. Now it should be clearer that we used the global full soil depth data which is listed in the data description. L255: Yes, it is added into the Section 3.1. L263: revised. L264: revised. L266: Please see the updated Table 2. L272: We have changed according to the suggestions. Now the sections on the spatial distribution of soil carbon is merged and a new section is added below on vegetation and GPP. L310: Added. Please see the updated Section 4.5 and Figure S5. L312: Revised. L328-329: Yes, as I mentioned in the manuscript, pattern maintains but the contribution is dampened when we used larger soil depth. L331-332: Revised. I found the numbers in the abstract is derived from an old experiment which is now corrected. L349: revised. L369: revised L368-370: Yes, revised according the previous similar suggestions. Now it is in the method. Also I added a paragraph of discussion on Cveg in Section 5.1. L375: revised L384-385: This is actually a part of the method where we extrapolated the soil and the results in shown in the supplement. We put it in the supplement not in the results section because it is too technical for the broad audience. We would like to leave it this way if it is possible. L386: revised. L388: Yes, we are aware of that. L388-389: revised. L396-403: It is a very good question. Unfortunately, we don't know the globally distribution of the metabolically active soil depth. The only thing we can do right now is to fit the vertical soil profile using statistical approach and that is what we did. But better understanding on this matter is truly important, however, beyond our ability to answer. L405-409: Yes, added in Section 5.2. L410: revised. L418-420: noted. L424-425: revised. L427: revised. Reference: revised and added.

Sincerely, Naixin Fan, on behalf of the co-authors.

Please also note the supplement to this comment:
https://www.earth-syst-sci-data-discuss.net/essd-2019-235/essd-2019-235-AC1-supplement.pdf

––––––––––––––––––––––––––––––

[Figure]

**Supplement:**

**Response to the referee comments**

Dear referees:

Thank you for making these constructive comments on our manuscript. The responses to your comments can be found below. Please also take note on the marked-up manuscript in the following pages after the response letter, where all the changes from the original submission are highlighted.

RC1 (sentences in blue colour is the original comments from the reviewer and the answer is in black colour):

I would be interested to hear more information about some of the derived datasets. For example, the creation of the herbaceous carbon stock map is described but what is the relative proportion of vegetation carbon found within the herbaceous layer is not stated? As the GPP ensemble is used in the estimation of the herbaceous layer what is the uncertainty in the herbaceous carbon content? How does the herbaceous carbon stock influence ecosystem turnover time vary in space, i.e. could it have been neglected?

Answer: We added some analysis of herbaceous biomass in the text per request. Please see the updated manuscript in detail but the simple answer is that the herbaceous biomass plays a minor role in the estimation of $\tau$ since it is less than 1% of soil carbon stock and 5% of vegetation carbon stock.

Similarly, the soil carbon estimated to maximum depth would be interesting to investigate further. A really simple but nice addition would be a map of the maximum soil depths inferred by your analysis.

Answer: We added the soil depth global distribution map (Figure S6). Please also note that the full soil depth is not inferred by our analysis but a global dataset (see Method).

The current text is a little unbalanced towards Csoil sometimes to the exclusion of Cveg or GPP in the introduction, results and discussion sections. The introduction sets out the overall challenge and usefulness of such datasets in constraining Earth System Models and their role in quantifying the response of the ter- restrial ecosystem to climate change. However, the fact that this is an update paper is not made fully clear. Doing so would I think make it straight forward to highlight the weaknesses of the previous analysis and how they are being improved here making a more robust and unique dataset. I honestly do support making updates and im- provement to existing datasets as this provides a clear traceable advancement in the science. Because the current manuscript does not clearly highlight soil as a weakness / uncertainty of existing works the introduction reads as being very soil dominated with little introduction of the vegetation carbon stock challenges or the estimation of GPP.

Answer: Yes, we updated the manuscript to a more balanced way. Please see details in the line to line answer below and the marked-up manuscript.

The introduction does clearly state one of the key assumptions, that ecosystems are assumed to be in steady state. What is missing is an appreciation that much of the worlds vegetation is not in steady state, either due to direct human intervention (biomass removal or other land use change) or as a result of increasing CO2 con- centration. Attempting to quantify this is out of scope but I think it would be useful to include either in the introduction or discussion the potential implications of this assump- tion leading to an underestimate in turnover times (e.g. Ge et al., 2018).

Answer: Yes, we updated the manuscript to address this matter. Please see details in the line to line answer below and the marked-up manuscript.

The results section, like the introduction, seems to be biased towards soil carbon re- sults rather than a complete overview. This should be addressed. Further information can be found below in the technical comments. The discussion lacks any discussion of the vegetation carbon stocks and almost any discussion of the GPP estimates. I also find it odd that figures 1-4 are not mentioned in the discussion at all. The discussion lacks sufficient comparison with existing studies / ESM outputs which this dataset should be constraining. One exception being the comparison with Todd-Brown et al., 2013 comparing soil carbon turnover times from CMIP5 models. Discussion of GPP importance is limited to its uncertainty contribution in the current analysis. While I have no problem with your choice to use FLUXCOM GPP estimates as observation-orientated. I do think it would be useful to include some discussion / context that compares your GPP estimate to alternate approaches e.g. remote sensing products (e.g. Zhang et al., 2017) or terrestrial ecosystem models constrained with remote sensing (e.g. Norton et al., 2019).

Answer: Yes, we updated the manuscript to a more balanced way. Please see details in the line to line answer below and the marked-up manuscript.

Response to the technical comments:

Please check the modified manuscript if the answer after the number of lines is 'revised'. Otherwise, the specific modifications or reasons are listed.

L14: revised.

L14-15: revised.

L16: revised.

L19: the interquartile range.

L19: added the reference.

L22: revised.

L22: Actually, the supporting results are shown in section 4.5 and Figure 4. Yes, this is probably a little bit surprising that Cveg doesn't contribute much to the global uncertainty. This is because the uncertainty in soil carbon stock is dominant.

L24: Added reference.

L29-32: Great suggestions! The regarding part is moved up.

L37: revised.

L39: Indeed, revised.

L41-43: They are added in the discussion. Please see Section 5.2.

L49-55: revised.

L81: revised.

L100: revised.

L108: revised.

L112: 'PH' is deleted because of not related to the context.

L167: revised.

L175-180: Yes, it is added in Section 3.1.

L214: No, but we propagated the uncertainty of different vegetation datasets into the turnover estimations.

L220: revised.

L223-234: revised. Now it should be clearer that we used the global full soil depth data which is listed in the data description.

L255: Yes, it is added into the Section 3.1.

90   L263: revised.

L264: revised.

L266: Please see the updated Table 2.

L272: We have changed according to the suggestions. Now the sections on the spatial distribution of soil carbon is merged and a new section is added below on vegetation and GPP.

95   L310: Added. Please see the updated Section 4.5 and Figure S5.

L312: Revised.

L328-329: Yes, as I mentioned in the manuscript, pattern maintains but the contribution is dampened when we used larger soil depth.

L331-332: Revised. I found the numbers in the abstract is derived from an old experiment which is now corrected.

100   L349: revised.

L369: revised

L368-370: Yes, revised according the previous similar suggestions. Now it is in the method. Also I added a paragraph of discussion on Cveg in Section 5.1.

L375: revised

105   L384-385: This is actually a part of the method where we extrapolated the soil and the results in shown in the supplement. We put it in the supplement not in the results section because it is too technical for the broad audience. We would like to leave it this way if it is possible.

L386: revised.

L388: Yes, we are aware of that.

110   L388-389: revised.

L396-403: It is a very good question. Unfortunately, we don't know the globally distribution of the metabolically active soil depth. The only thing we can do right now is to fit the vertical soil profile using statistical approach and that is what we did. But better understanding on this matter is truly important, however, beyond our ability to answer.

115   L405-409: Yes, added in Section 5.2.

L410: revised.

L418-420: noted.

L424-425: revised.

L427: revised.

120  Reference: revised and added.

**Apparent ecosystem carbon turnover time: uncertainties and robust features**

125  Naixin Fan[1], Sujan Koirala[1], Markus Reichstein[1], Martin Thurner[3], Valerio Avitabile[4], Maurizio Santoro[5], Bernhard Ahrens[1], Ulrich Weber[1], Nuno Carvalhais[1,2]

[1]Max Planck Institute for Biogeochemistry, Hans Knöll Strasse 10, 07745 Jena, Germany

[2]Departamento de Ciências e Engenharia do Ambiente, DCEA, Faculdade de Ciências e Tecnologia, FCT, Universidade Nova de Lisboa, 2829-516 Caparica, Portugal

130  [3]Biodiversity and Climate Research Centre (BiK-F), Senckenberg Gesellschaft für Naturforschung, Senckenberganlage 25, 60325 Frankfurt am Main, Germany

[4]European Commission, Joint Research Centre, Via E. Fermi 2749, 21027 Ispra, Italy

[5]Gamma Remote Sensing, 3073 Gümligen, Switzerland

135  *Correspondence to*: Naixin Fan (nfan@bgc-jena.mpg.de) and Nuno Carvalhais (ncarval@bgc-jena.mpg.de)

**Abstract.** The turnover time of terrestrial carbon ($\tau$) is an emergent ecosystem property that quantifies the strength of global carbon cycle – climate feedback. However, observations and simulations of the magnitude of $\tau$ and its response to climate change is still characterized by large uncertainty. In this study, by assessing apparent carbon turnover time as the ratio between carbon stocks and fluxes, we provide an update of diagnostic terrestrial carbon turnover times estimations and

140  associated uncertainties on a global scale using multiple, state-of-the-art, observation-based datasets of soil organic carbon stock ($C_{soil}$), vegetation biomass ($C_{veg}$) and gross primary productivity (GPP). In spite of the large uncertainties in the different $\tau$ estimation, our findings reveal that the latitudinal gradients of $\tau$ are consistent across different datasets and soil depth. Furthermore, there is a strong consensus on the negative correlation between $\tau$ and temperature along latitude that is stronger in temperate zones (30ºN-60ºN) than in subtropical and tropical zones (30ºS-30ºN). Using this new ensemble of

155 data, we estimated the global average $\tau$ to be $40^{+7}_{-11}$ (median ± interquartile) years when the full soil depth (usually the soil depth beyond 100cm, see Methods) 
[revised manuscript text omitted]
 which is usually higher than 100cm. In permafrost soil, full soil depth can extend beyond 400cm (Figure S6).

**2.4 The FLUXCOM global gross primary productivity dataset**

FLUXCOM is an initiative to upscale biosphere-atmosphere fluxes measurements from eddy covariance flux towers (FLUXNET) to global scale (Jung et al., 2017). In this study, we used the mean annual GPP datasets based on remote-sensing forcing and nine machine learning methods with two flux partitioning methods trained on daily carbon fluxes, that is, 18 members of GPP (Tramontana et al., 2016). In order to produce high resolution (0.083º) spatial grids of carbon fluxes, only high-resolution satellite-based predictors were used in model training. In this study, we derived the long-term mean

350 annual GPP by averaging annual GPP from 2001 to 2014. We note that all the 18 members is used independently to estimated τ (not averaged).

**2.5 Climate datasets**

[revised manuscript text omitted]

**4.4 The vertical distribution of global carbon stock**¶

**4.4 The spatial distribution of vegetation and GPP**

In comparison with soil carbon, the results show much more consistency and convergent global number of carbon stock among the four global vegetation datasets (Figure 3). Our results show that global vegetation carbon stock is 10% to 25% of the global soil carbon stock, depending on soil depth. The high spatial correlations (r>0.75) between each pair of data indicate the current estimations of vegetation is consistent. Different from the spatial distribution of soil carbon, most vegetation carbon is located in the tropics whereas much less carbon in higher latitudes. As a matter of fact, the $C_{veg}$ in circumpolar region is only 10% of that in non-circumpolar region (Table 2). Here we address that the $C_{veg}$ is consist of three components including AGB, BGB and herbaceous biomass among which the herbaceous biomass is estimated from mean annual GPP (see Methods). We show the herbaceous biomass is only 5% of the total $C_{veg}$ and less than 1% of the total $C_{soil}$ indicating the minor role of herbaceous biomass in affecting the spatial distribution of total carbon stock and the uncertainty. The comparison among the four vegetation datasets shows relatively higher level of disagreement in arid and some cold regions (Figure 3, upper off-diagonal subplots). Nevertheless, the current estimations of global vegetation from different sources show consistent spatial distributions.

Our results show that the spatial pattern of the global GPP is similar to the $C_{veg}$ where there is higher primary productivity in the tropics and lower in the higher latitudes (Figure 4). The different members of the GPP estimation (see Methods) show very high consistency globally except for arid and polar region. The relative uncertainties in arid and polar region range from 50% to 100% whereas there is less than 50% of uncertainties in other regions.

Although the differences among different vegetation and GPP estimations, in general, are not as high as soil, we show the uncertainties can be regionally high. We thus next investigate the contribution of each component to the spatial distribution and uncertainty.

**4.5 The ecosystem carbon turnover times and associated uncertainties**

Using $C_{soil}$, $C_{veg}$ and GPP, we estimated the carbon turnover times with different combinations of datasets in order to quantify the uncertainty. We calculated τ in the same manner as the previous study (Carvalhais et al., 2014) in which they used only 1m of soil in the circumpolar region and full soil depth in the non-circumpolar region. We compared the spatial distribution of the τ estimations of the previous and our study (Figure S5). Our results show a large range of relative difference and low spatial correlation (r = 0.51). We found the main differences are in the northern circumpolar region, which is caused by the differences in the $C_{soil}$ estimations, indicating a large uncertainty of τ estimations the in this region. Our estimation of global mean τ is 35 years with an interquartile range of 29 to 43 years, which is much longer than the previous study of 23 years with interquartile range of 19 to 30 years. In addition, we derived a global τ of 40 years with an interquartile range of 29 to 47 years by assuming the maximum active layer thickness to be the full soil depth in the circumpolar regions instead of using only 1-meter $C_{soil}$ as was done in the previous study. The incorporation of deep soil in the circumpolar region increased the global τ by 7 years. The global spatial distribution of τ (Figure 5) shows great

[revised manuscript text omitted]

The correlation between $\tau$ and precipitation, in general, has larger variability across latitude and a higher uncertainty due to differences in data (Figure 8b). Contrary to the $\tau$ - T relationship, the uncertainty of the $\tau$ - P relationship derived from both different data sources and soil depths are smaller in the tropics than in high latitudes. Negative correlations dominate the latitudes between 20 and 50º N as well as between 20 and 40º S, while there is a stronger positive correlation in the tropics. There is a shift in the sign of the correlation coefficient from negative in temperate zone to positive in tropics, indicating the role of water changes from water-limited regions to water-excessive regions. We found the pattern of correlation between $\tau$ and precipitation is different from the previous study (Carvalhais et al., 2014), especially in the tropics. We therefore investigated the possible cause of the difference by mixing all components ($C_{soil}$, $C_{veg}$ and GPP) between the previous and current study. By examining the correlation between each mixed $\tau$ estimation and climate factors, our results show that positive correlation in the tropics is caused by the $C_{soil}$ (Figure S7). This is consistent with the previous results (Figure 1) which shows large difference in the spatial distribution of $C_{soil}$ in the tropics between the three soil datasets we used in this study and HWSD soil dataset.

**5 Discussion**

In this section, we will discuss the robustness of the current state-of-the-art estimation on global terrestrial carbon turnover times and their response to climate change. We first show the variation of spatial and vertical distribution of carbon stock in different regions and the possible reason for the difference, and we then discuss the robustness of zonal distribution of turnover times and zonal changing rates across different datasets. Finally, we focus on the sensitivity of turnover times to climate and implications.

**5.1 Estimation of global carbon stock**

Accurate estimation of terrestrial carbon storage and turnover time are essential for understanding carbon cycle-climate feedback (Saatchi et al., 2011; Jobbágy et al., 2000). Our analysis benchmarks soil, vegetation carbon storage and GPP from multiple state-of-the-art observational based datasets at global scale and provide not only an estimate of the total carbon stock but also the vertical distribution and spatial variability of global carbon stock. We divide the global map into circumpolar and non-circumpolar regions due to the different characteristics and uncertainty.

We found that there is a significant difference across the current soil carbon datasets in both circumpolar and non-circumpolar regions. The results show that the uncertainty of $C_{soil}$ estimations in the circumpolar region is two times larger than that of the non-circumpolar region (Figure 6). The spatial patterns of total ecosystem $C_{soil}$ among the soil datasets are

610 more consistent in the non-circumpolar region than in the circumpolar region. In contrast with the non-circumpolar region, there is lower confidence in the circumpolar region in estimating $C_{soil}$ due the fact that there is low spatial correlation across datasets. The difference can be caused by various reasons. As an important input to the machine learning method, in-situ soil profiles are very important factors that influence the final results of the upscaling. The sparse coverage of soil profiles in the circumpolar region may cause the large divergence in the northern circumpolar region. A major difference between S2017

615 and the other two soil datasets is that soil carbon stock was a direct target of upscaling in the former dataset, while in the latter two datasets each component used to calculate $C_{soil}$ (carbon density, bulk density and percentage of coarse fragments) was predicted individually. In addition, the climatic covariates that were used in the upscaling were different (see Method).

In contrast with the non-circumpolar region, the circumpolar $C_{soil}$ does not have a decreasing trend up to 4 meters of soil depth (Figure S1) which indicates that there is a significant amount of carbon stores in deep soil. The deep soil turnover is a

620 key process to the global carbon cycle yet poorly understood (Todd-Brown et al., 2013). In this study, we extrapolated the soil carbon stock to full soil depth. We chose the model ensembles from a framework to pick out the models that had a minimum distance between prediction and observations by using in-situ soil profiles (see Supplement). Two model ensembles were selected that can best represent the soil vertical distribution in circumpolar and non-circumpolar regions by comparing model simulations and in-situ observations. The final results depend on the information from the soil profiles and

625 also the characteristics of the empirical models. The extrapolation gave us insights to the carbon storage and vertical distribution in deep soil. The results of extrapolation show there is approximately 15% of carbon stored below 2 meters globally and over 20% of carbon stored below 2 meters in the northern circumpolar region. Although the total amount of carbon storage in the ecosystem shows a large divergence among different datasets, the ratio between different soil depths are quite consistent indicating a high confidence in the vertical structure of soil compare to the total amount.

630 The global soil carbon stocks across observational-based datasets are much less divergent than the current earth system model (ESM) simulations. The CMIP5 results show the simulated carbon storage ranges from 500-3000 PgC making τ varies by a factor of 3.6, from 11 to 39 years (Todd-Brown et al., 2013). Our results show that the amount of carbon in the ecosystem is much higher than the estimation by ESMs. Even the lowest estimation (S2017 dataset) of total carbon storage is about 500 PgC higher than the highest ESM estimation (MPI-ESM-LR). The spatial distribution of carbon stocks among

635 ESMs have a large variation across models while the observational-based datasets are more consistent in the non-circumpolar region. But we leave a question mark to the soil carbon in the circumpolar region, which is characterized by large uncertainty as shown by the current observational-based soil datasets.

Compared with soil, the higher level of consistency in the vegetation and GPP estimates indicate there is a consensus on the current estimations in the above-ground carbon stock. However, we note that the regional differences in the products can

640 significantly affect the spatial distribution and uncertainty of τ. Nevertheless, vegetation and GPP contributes a little to the global mean value of τ estimations.

**5.2 The terrestrial carbon turnover time and uncertainty**

The uncertainty analysis showed that our current estimation of $\tau$ has a considerable spread which derived from the state-of-the-art observations of soil, vegetation and carbon fluxes. In this study, we showed the uncertainty is contributed mainly by the soil carbon stock and GPP, where the former dominates the vast areas in the circumpolar region and the tropical peatland while the latter dominates the semi-arid and arid regions. We showed that GPP is the second largest contributor to the total uncertainty which potentially leads to significant differences in the estimation of $\tau$ considering different products of GPP (Zhang et al., 2017; Norton et al., 2019). This result is consistent with the previous study (Todd-Brown et al., 2013) that the bias in estimated primary productivity can affect the carbon turnover estimations to a large extent not only by using observational-based data but in the ESMs simulations. However, the uncertainty comes not only from the differences across datasets but also from the soil depth we chose to estimate $\tau$. The frozen permafrost soil in the circumpolar region, although containing a large amount of carbon is an important component in the process of turnover (Zimov et al., 2006). However, we do not know to what soil depth we should use in the $\tau$ estimation since currently our knowledge on the active layer thickness of frozen permafrost soil is still lacking. In addition, the active layer thickness of permafrost changes with climate, which adds more uncertainty to the estimation of $\tau$. Thus, we argue that the current datasets cannot support robust estimation of global $\tau$. It is worth to note that our estimation of $\tau$ is based on the steady-state assumption, that is, the net exchange of carbon between the terrestrial ecosystem and the atmosphere equals to zero. In our study, the steady-state assumption is a proper assumption for that our analysis focused on the $\tau$ estimation at long-term temporal scale and large spatial scale. However, this assumption is valid to a much less extent at site-level as the net exchange of carbon is, most of the time, not in balance (Ge et al., 2018).

Although the current estimation of $\tau$ has a large variation, we show that the zonal distribution of $\tau$ is a robust feature that changes little with different datasets, which indicates that the current state-of-the-art datasets all agree on the latitudinal gradient of the carbon turnover time. Another robust feature is that the zonal changing rate of $\tau$ does not change with the soil depth (Figure 7). It has always been a problem of what soil depth should we use to represent the functional part of carbon in the ecosystem. The selection of soil depth is usually arbitrary and varies from study to study. For example, Koven et al. (2017) and Wang et al. (2017) used the top one-meter of soil carbon to represent the total terrestrial carbon pool while Carvalhais et al. (2014) extrapolated soil to full depth and used it as the pool. Our results demonstrate that the selection of the soil depth does not affect the zonal pattern that we observed. This can be better seen in the next section with the response of $\tau$ to climate.

**5.3 Robust associations of $\tau$ and climate**

Despite the large uncertainty in the $\tau$ estimations, we identified robust response of $\tau$ to climate change. It is well recognized that the sensitivity of terrestrial carbon to climate is a major uncertainty, which is reflected by the spread of $\tau$ estimation by the different ESMs. However, we need reliable estimations of $\tau$ to quantify its climate sensitivity and provide robust constraints to improve the performance of the current ESMs. We showed the zonal correlation between $\tau$ and temperature varies with latitude where high correlations are found in the high latitude and low to moderate correlation in low latitude,

especially the tropics. The zonal pattern of τ-precipitation is more complicated in that water availability can cause local variability to a great extent. The correlation between turnover times and precipitation in the tropics is higher than that with temperature as shown in Figure 8d indicating a potentially more dominant role of precipitation in the tropics. The role shifted along latitude between temperature and precipitation in the pattern of τ due to the variation in the relative importance for each parameter. However, the temperature gradient shaped the zonal distribution of τ as it can be seen that τ increases with latitude. All of these relationships are verified by each ensemble member of the data. We found the correlations, although they vary in strength, are very robust. The intimate interaction of energy and water along with other factors such as land use change all affect τ but on different spatial and temporal scales. It is worth mentioning that the τ - T relationship is similar when compared with previous results (Carvalhais et al., 2014) whereas there are considerable differences in the τ - precipitation relationship, specifically in tropical regions where the turnover times were always negatively correlated with precipitation in previous study. The different τ – P zonal patterns of correlation between the previous and the current study, as shown before, is mainly caused by the difference in the soil carbon stock (Figure S7). This finding indicates the response of τ to moisture is characterized by large uncertainty.

[revised manuscript text omitted]

**Supplementary material**

**1 Mass conservative aggregation**

In order to unify the spatial resolution and geographic coordinate system of dataset from different sources, we need to make sure that the total amount of stock for soil, vegetation, etc. doesn't change during aggregation and transformation, i.e. the variable need to be mass conservative. However, 'state' variable such as temperature, vegetation types do not need to fulfill the mass conservative requirement, nor they should. In our study, we developed a mass conservative method to maintain mass for carbon stocks. We first multiply the variable that need to be aggregated ($X_{fine}$) by corresponding land area ($A_{fine}$) at grid cell level represented by equation (1), then aggregate the product ($XA_{fine}$) by summing the values in N×N grids cell depending on the target resolution (equation (2)). The land area is also sum to the target resolution (equation (3)). Finally, the area-weighted variable is derived by dividing aggregated product ($XA_{coarse}$) by corresponding land area ($A_{coarse}$) as illustrated by equation (4).

$$XA_{fine} = X_{fine} \times A_{fine} \quad (1)$$

$$XA_{coarse} = SUM(XA_{fine}) \quad (2)$$

$$A_{coarse} = SUM(A_{fine}) \quad (3)$$

$$XA_{coarse} = \frac{XA_{coarse}}{A_{coarse}} \quad (4)$$

We applied the method to all datasets that requires aggregation including soil, vegetation and GPP that were used in the study.

**2 Bulk density correction**

The bulk density (BD) in SoilGrids and LandGIS are too high due to two reasons. First, the measurements of BD are less and missing in many horizons (Hengl et al., 2017). And the measurements of BD in permafrost region, especially in Canada forest soil and Russian, are problematic (personal communication with Tomislav Hengl). In this study, we applied a pedotransfer function from Köchy et al. (2015) to make correction based on organic carbon concentration (we only applied the function to the grid cells where carbon > 8%):

$$BD = (1.38 - 0.31 \times \log\left(\frac{OC}{10}\right)) \times 1000$$

**3 Model selection for extrapolation of soil**

In this section, we introduce the framework that we used to select the models for extrapolating soil from $0 - 2m$ to full soil depth.

**3.1 Different characteristics of permafrost and non-permafrost soil**

The amount and vertical distribution of soil organic carbon are largely influenced by vegetation which fixes atmospheric $CO_2$ and transport carbon into the land ecosystem. However, the SOC stock have a much more complicated relationship with productivity of plants than a simple linearly one (Jackson et al., 2017). The higher biomass, which implies more carbon sequestration by aboveground biomass, however, does not necessarily lead to increases in SOC storage. Although the processes of soil formation, accumulation, and stabilization have been intensively studied and debated, the mechanisms that determine the soil carbon stock, especially in deeper soil, are still unclear. Instead of using process modelling approach, we chose statistical approach to extrapolate each soil profiles in the gridded dataset from 2m to full depth. The reason of performing soil carbon stock extrapolation is that we have little knowledge on how much the carbon stored in the soil that is deeper than 2m, although deeper soil is a crucial component in the climate-carbon cycle feedback. The other reason is the different dataset report SOC stock at different depths. The advantage of using statistical method is that we do not need to know the mechanisms that control the soil processes. Instead, we select simple empirical mathematical models that can represent and predict the in-situ soil profiles.

We used 425 permafrost peatland profiles from ISCN soil database and 1000 profiles from WOSIS soil database to study the characteristics of vertical distribution of SOC. Figure S1 shows the accumulated SOC stock profiles with depths in permafrost and non-permafrost region. The vertical distribution of carbon with depth in permafrost soil has a distinguished feature that the SOC has a high linear relationship with depth. This fact implies the soil carbon keeps increasing even after 3 meters in permafrost soil (Figure S1b). However, we have no idea to what depth can soil carbon keep increasing and the total amount of the storage in permafrost peatland due to the limited observational depth of SOC. In contrast, soil profiles in non-permafrost region stop increasing mostly before 2 meters. The results demonstrate the necessity of extrapolating soil to full depth, especially for permafrost soil.

**2.2 Selection of models**

We included 12 models (Table S1) for predicting SOC stock to full soil depth. Figure S2 shows an example result in which the data points that is shallower than 1m were used to fit all the models and predict the point that is deeper than 2m for a

typical soil profile. Due to the different mathematical characteristics of the models, the prediction has quite a spread. Relatively 'conservative' models including model ensemble BHIJKL tend to underestimate the carbon stock while the more 'aggressive' ones ACDEFG tend to overestimate the stock.

1000 In the sense that we do not know which one or group of models can best predict the accumulated carbon storage, we conducted a selection process (see Methods) by grouping all the models into all possible combination and rank the performance for all the model averaging results as shown in Table 2. All the models were used to fit the WOSIS data which covers most of the biomes and ISCN database which covers only permafrost soil. We conducted three batch of experiments in the same manner but used data points within different depths. The data points lower than 50cm, 100cm and 200cm were

1005 used to predict the SOC that is higher than 200cm. Our goal is to find the ensemble of models that has the highest model performance, the best coverage, the minimum error and AIC.

**2.3 Extrapolating soil with different method**

The main goal of using several models is to search for the best group of models that can best predict the vertical distribution

1010 of soil carbon stock and we compared the below approaches for that purpose:

1. The Bayesian Model Averaging (BMA) method is used in this study to find the best model ensemble for the prediction of soil carbon storage to full depth. The MODELAVG Matlab toolbox (Vrugt, 2016) which implemented many different model averaging techniques including BMA method. The advantage of BMA method is that it considers explicitly the uncertainty of prediction of a target variable which can provide a probabilistic distribution of weight for each model

1015 instead of only a weighted-average, deterministic prediction. By maximize the likelihood function from the training dataset, the weights $\beta = \{\beta1, …,\beta k\}$ and standard deviation $\sigma = \{\sigma1, …, \sigma k\}$ are estimated.

2. Equal weights averaging (EWA) which consider each the participating model have the same weight and the prediction is derived by equal-weighted averaging the model results.

1020 The complete combination among different models are also compared and the best model ensembles are obtained by maximizes MEF, minimizes KL and minimizes AIC. The results show that EWA and BMA methods have similar performances (Table S2). We choose EWA method due to it have a slightly better coverage of observations.

Two model ensembles were selected from the model selection framework that can best represent circumpolar and non-circumpolar region based on observational datasets in the two regions, respectively. The performance of the chosen

1025 ensemble is synthesized in Figure S3. It shows the ensemble DIJKL overall can well predict the carbon stock in non-circumpolar region that is deeper than 200cm only using points lower than 50cm. Model efficiency is 0.83 and the residue between observation and prediction is little biasd in the prediction (Figure S3c). The histogram (Figure S3b) shows the prediction has the same distribution as the observations. The ensemble was also used to predict observations within different percentiles and over 70% of observations can be included in the uncertainty ($[-\sigma, +\sigma]$). The results show that the selected

1030 models can well represent the vertical distribution of $C_{soil}$ thus we used them to extrapolate the global gridded datasets in order to obtain the total soil carbon storage in the soil. The selected model ensemble ACDEF for the circumpolar soil have lower model efficiency and less well represent the soil in the region (Figure S4). We then applied extrapolation on three global datasets which are Sanderman, SoilGrids and LandGIS. The averaged results of ensemble DIJKL is used to extrapolate non-circumpolar soil from 2m to full soil depth and ensemble ACDEF to extrapolate circumpolar soil.

1045

[Figure]

Figure S9: **The vertical distribution of accumulated SOC stock (kg.m-2) with depth (cm).** (a) 425 soil profiles of permafrost peatland region and (c)1000 soil profiles of non-permafrost region. The probability distribution density of SOC for (b) permafrost, (d) non-permafrost. The blue open circle represents observational data points in each profile.

1050

1055

[Figure]

Figure S10: **An example of soil profile vs models.** Overlay the observational points and model results.

1060

[Figure]

**Figure S11: Performance of the averaged results of model D, I, J, K and L in predicting soil carbon storage from 50cm to 200cm using WOSIS data.** (a) Ensemble mean vs. observation, 1:1 line in blue. (b) The histogram of observation, model ensemble and each model. It shows the Kullback-Leibler distance from model ensemble mean to observation, the two-sample Kolmogorov-Smirnov test (1 represent the model ensemble mean and the observation come from the same distribution, 0 otherwise), the p-value of Kruskal-Wallis test (significant if p<0.05). (c) residue between model ensemble mean and observation. KS represents the one-sample Kolmogorov-Smirnov test (1 represent the model ensemble mean and the observation come from the same distribution, 0 otherwise). AD represents Anderson-Darling test (1 represent the model ensemble mean and the observation come from the same distribution, 0 otherwise). (d) The coverage of observation data points within [-σ, +σ], [min, max], [25%, 75%] and average.

[Figure]

**Figure S12: The same as Figure 3 except for using ISCN data and model ensemble of A, C, D, E and F to predict soil carbon storage from 200cm to deep soil.**

[Figure]

**Figure S13: Comparison of τ estimations between the previous study (Carvarhais et al., 2014) and the current study.** The upper off-diagonal subplot is the ratios between each pair of datasets (column/row). The bottom off-diagonal subplot shows the major axis regression between each pair of datasets (m: slope, b: intercept, r: correlation coefficient).

[Figure]

**Figure S14: Global distribution of full soil depth.**

[Figure]

1085

**Figure S15: The zonal pattern of the correlation between τ and climate factors.** Each component ($C_{soil}$, $C_{veg}$ and GPP) from the previous study (Carvalhais et at., 2014) is mixed with each component of the current study. The prefix 'old' stands for the component from the previous study and the prefix 'new' stands for the component from the current study.

1090

**Table S1:** Empirical functions candidates for extrapolation of soil carbon

| | Equation |
|---|---|
| A | $a \cdot D^b + C$ |
| B | $a \cdot e^{b \cdot D} + c \cdot e^{d \cdot D}$ |
| C | $a \cdot log\,(b \cdot D + 1)$ |
| D | $a \cdot log\,(b \cdot D + c)$ |
| E | $K \cdot log_{10}(D) + I$ |
| F | $(10^I \cdot D^{K+1})/(K+1) + c$ |
| G | $a + b \cdot D$ |
| H | $b \cdot (1 - \beta^D)$ |
| I | $b \cdot (1 - \beta^D)^a$ |
| J | $a \cdot (1 - e^{-(D/b)^c})$ |
| K | $a \cdot (1 - e^{-b \cdot D})^c$ |
| L | $a \cdot (1 - \dfrac{log(1 - (1-b) \cdot e^{-c \cdot D})}{log\,(b)})$ |

1095

1100

**Table S2: Performance of different methods.**

| | Circumpolar | | Non-circumpolar | |
|---|---|---|---|---|
| | **EWA** | **BMA** | **EWA** | **BMA** |
| **RMSE** | 36.575 | 37.686 | 5.482 | 5.292 |
| **AIC** | 1516.134 | 1528.526 | 3977.226 | 3895.513 |
| **KL** | 0.020 | 0.020 | 0.039 | 0.036 |
| **MEF** | 0.640 | 0.617 | 0.862 | 0.872 |
| **Coverage (%)** | 14.5 | 13.5 | 61.9 | 50.3 |

---

## Author Comment (AC2) · 31 May 2020

Dear referees:

Thank you for making these constructive comments on our manuscript. The responses to your comments can be found below. Please also find the marked-up manuscript in the attachment, where all the changes from the original submission are highlighted. In the following response, I first list your original question or comment then response to it.

Responses to the comments:

1) The dataset can only be downloaded when the users registered on the website. After I registered, somehow, I still cannot download the dataset. So, I only reviewed the manuscript not the dataset. Whether the original data and the process data used to

derive the turnover time can also be downloaded from the link? This would be helpful for people trying to reproduce the data generation process or for those that would like to use original data or process data. Answer: We have checked the issue and the link seems working fine to us. We received data downloading request from people outside of our institute and they are able to obtain the data. Maybe a second attempt could solve the problem? Anyway, please inform us if there is still problem downloading the data. We would love to help. 2) The turnover time was estimated assuming steady state, in which the efflux equals to the influx. While the reality is in non-steady state. The effects of this assumption on the estimation of turnover time should be discussed. Answer: Yes, please see Section 5.2 in the updated manuscript. 3) Was the high consistency of vertical structure of soil carbon storage caused by the consistent extrapolation model? i.e. same model parameters lead to the same vertical ratio? (P15L393) Answer: No, our empirical models extrapolate soil to the full soil depth. But if one looks at the vertical structure before 2 meters (that is the maximum provided depth of the three soil datasets), They are also similar (Table 2). 4) How to compare the sensitivities of turnover times to precipitation and temperature? They have different units (P16L430). Answer: Thanks for the question! We rephrased to a more accurate statement. Please see the updated line. 5) The influence of other factors on turnover times are missing. Could you give further results or discussion? (P16L435) Answer: Our focus in this study is the contribution of uncertainty from different components. And we tried to see if the established pattern of latitudinal correlation between turnover and climate factors from the previous study (Carvalhais et al., 2014) is robust using our updated estimations. The effect of other factors worth another paper to address therefore we prefer not to involve in this paper.

6) The GPP only used one data source, i.e. FLUXCOM produced by Jung. There are also other sources of GPP such as the GPP generated using LUE model published in Nature Scientific Data. It would be interesting to see the change in uncertainty. Answer: Please see the added discussion on this matter in Section 5.2.

[Figure]

Specific comments: P6L188: The R and r is not consistent. L188: revised. P10L256: The vegetation biomass is missing in the first sentence. L256: revised. P14L378: "caused" should be "caused by". L378: revised. P16L436: Why the relationship between turnover time and precipitation are different with previous studies? L436: It is because of the large difference between new and previous estimations of soil. Results and discussion on this matter is added into the manuscript (also see Figure S7).

P16L447: Typo. Should be "state-of-the-art". L447: revised. P19L570: The color of this reference is different from other parts of the manuscript. Fig. 1 and Fig. 2: It should be noted that the bottom diagonal subplot was the regression of row with column, i.e. y=row, x=column? Besides, what did the color around the origin represent? L570: The color of the reference is adjusted. More description of the Figure 1 is added. The color around the origin is the density of the data which is also specified in the caption. Fig. 3: Quantile range here is 25Fig. 5: How to determine the turning point? It seems like not 0? Fig. 5: We try to locate the local maximum by searching the latitudinal turnover values. You are right, the point is actually is little bit below 0. Fig. 6: The lines in subplot c and f indicate? The individual lines in Fig. 6 is each member of the turnover estimation. Terminology: The soil dataset provided by Sanderman et al 2017 was noted as S2017 in the text and the tables, while in the figures it was noted as Sanderman. Please be consistent through the manuscript. Revised. supplement-P2L32: CO2 should be CO2. supplement-P3L59: The period was missing between "Table 2" and "All". Revised.

Sincerely, Naixin Fan, on behalf of the co-authors

Please also note the supplement to this comment: https://www.earth-syst-sci-data-discuss.net/essd-2019-235/essd-2019-235-AC2-supplement.pdf

**Supplement:**

**Response to the referee comments**

Dear referees:

Thank you for making these constructive comments on our manuscript. The responses to your comments can be found below. Please also take note on the marked-up manuscript in the following pages after the response letter, where all the changes from the original submission are highlighted.

RC2 (sentences in blue colour is the original comments from the reviewer and the answer is in black colour):

1) The dataset can only be downloaded when the users registered on the website. After I registered, somehow, I still cannot download the dataset. So, I only reviewed the manuscript not the dataset. Whether the original data and the process data used to derive the turnover time can also be downloaded from the link? This would be helpful for people trying to reproduce the data generation process or for those that would like to use original data or process data.

Answer: We have checked the issue and the link seems working fine to us. We received data downloading request from people outside of our institute and they are able to obtain the data. Maybe a second attempt could solve the problem? Anyway, please inform us if there is still problem downloading the data. We would love to help.

2) The turnover time was estimated assuming steady state, in which the efflux equals to the influx. While the reality is in non-steady state. The effects of this assumption on the estimation of turnover time should be discussed.

Answer: Yes, please see Section 5.2 in the updated manuscript.

3) Was the high consistency of vertical structure of soil carbon storage caused by the consistent extrapolation model? i.e. same model parameters lead to the same vertical ratio? (P15L393)

Answer: No, our empirical models extrapolate soil to the full soil depth. But if one looks at the vertical structure before 2 meters (that is the maximum provided depth of the three soil datasets), They are also similar (Table 2).

4) How to compare the sensitivities of turnover times to precipitation and temperature? They have different units (P16L430).

Answer: Thanks for the question! We rephrased to a more accurate statement. Please see the updated line.

5) The influence of other factors on turnover times are missing. Could you give further results or discussion? (P16L435)

Answer: Our focus in this study is the contribution of uncertainty from different components. And we tried to see if the established pattern of latitudinal correlation between turnover and climate factors from the previous study (Carvalhais et al., 2014) is robust using our updated estimations. The effect of other factors worth another paper to address therefore we prefer not to involve in this paper.

6) The GPP only used one data source, i.e. FLUXCOM produced by Jung. There are also other sources of GPP such as the GPP generated using LUE model published in Nature Scientific Data. It would be interesting to see the change in uncertainty.

Answer: Please see the added discussion on this matter in Section 5.2.

Specific comments:

P6L188: The R and r is not consistent.

L188: revised.

P10L256: The vegetation biomass is missing in the first sentence.

L256: revised.

P14L378: "caused" should be "caused by".

L378: revised.

P16L436: Why the relationship between turnover time and precipitation are different with previous studies?

L436: It is because of the large difference between new and previous estimations of soil. Results and discussion on this matter is added into the manuscript (also see Figure S7).

P16L447: Typo. Should be "state-of-the-art".

L447: revised.

60    P19L570: The color of this reference is different from other parts of the manuscript. Fig. 1 and Fig. 2: It should be noted that the bottom diagonal subplot was the regression of row with column, i.e. y=row, x=column? Besides, what did the color around the origin represent?

L570: The color of the reference is adjusted. More description of the Figure 1 is added. The color around the origin is the density of the data which is also specified in the caption.

65    Fig. 3: Quantile range here is 25Fig. 5: How to determine the turning point? It seems like not 0?

Fig. 5: We try to locate the local maximum by searching the latitudinal turnover values. You are right, the point is actually is little bit below 0.

Fig. 6: The lines in subplot c and f indicate?

The individual lines in Fig. 6 is each member of the turnover estimation.

70    Terminology: The soil dataset provided by Sanderman et al 2017 was noted as S2017 in the text and the tables, while in the figures it was noted as Sanderman. Please be consistent through the manuscript.

Revised.

supplement-P2L32: $CO_2$ should be $CO_2$. supplement-P3L59: The period was missing between "Table 2" and "All".

75    Revised.

**Apparent ecosystem carbon turnover time: uncertainties and robust features**

80    Naixin Fan[1], Sujan Koirala[1], Markus Reichstein[1], Martin Thurner[3], Valerio Avitabile[4], Maurizio Santoro[5], Bernhard Ahrens[1], Ulrich Weber[1], Nuno Carvalhais[1,2]

[1]Max Planck Institute for Biogeochemistry, Hans Knöll Strasse 10, 07745 Jena, Germany

[2]Departamento de Ciências e Engenharia do Ambiente, DCEA, Faculdade de Ciências e Tecnologia, FCT, Universidade Nova de Lisboa, 2829-516 Caparica, Portugal

[3]Biodiversity and Climate Research Centre (BiK-F), Senckenberg Gesellschaft für Naturforschung, Senckenberganlage 25, 60325 Frankfurt am Main, Germany

[4]European Commission, Joint Research Centre, Via E. Fermi 2749, 21027 Ispra, Italy

[5]Gamma Remote Sensing, 3073 Gümligen, Switzerland

*Correspondence to*: Naixin Fan (nfan@bgc-jena.mpg.de) and Nuno Carvalhais (ncarval@bgc-jena.mpg.de)

**Abstract.** The turnover time of terrestrial carbon ($\tau$) is an emergent ecosystem property that quantifies the strength of global carbon cycle – climate feedback. However, observations and simulations of the magnitude of $\tau$ and its response to climate change is still characterized by large uncertainty. In this study, by assessing apparent carbon turnover time as the ratio between carbon stocks and fluxes, we provide an update of diagnostic terrestrial carbon turnover times estimations and associated uncertainties on a global scale using multiple, state-of-the-art, observation-based datasets of soil organic carbon stock ($C_{soil}$), vegetation biomass ($C_{veg}$) and gross primary productivity (GPP). In spite of the large uncertainties in the different $\tau$ estimation, our findings reveal that the latitudinal gradients of $\tau$ are consistent across different datasets and soil depth. Furthermore, there is a strong consensus on the negative correlation between $\tau$ and temperature along latitude that is stronger in temperate zones (30ºN-60ºN) than in subtropical and tropical zones (30ºS-30ºN). Using this new ensemble of data, we estimated the global average $\tau$ to be $40^{+7}_{-11}$ (median ± interquartile) years when the full soil depth (usually the soil depth beyond 100cm, see Methods) 
[revised manuscript text omitted]
 which is usually higher than 100cm. In permafrost soil, full soil depth can extend beyond 400cm (Figure S6).

**2.4 The FLUXCOM global gross primary productivity dataset**

FLUXCOM is an initiative to upscale biosphere-atmosphere fluxes measurements from eddy covariance flux towers (FLUXNET) to global scale (Jung et al., 2017). In this study, we used the mean annual GPP datasets based on remote-sensing forcing and nine machine learning methods with two flux partitioning methods trained on daily carbon fluxes, that is, 18 members of GPP (Tramontana et al., 2016). In order to produce high resolution (0.083º) spatial grids of carbon fluxes, only high-resolution satellite-based predictors were used in model training. In this study, we derived the long-term mean annual GPP by averaging annual GPP from 2001 to 2014. We note that all the 18 members is used independently to estimated $\tau$ (not averaged).

**2.5 Climate datasets**

[revised manuscript text omitted]

**4.4 The spatial distribution of vegetation and GPP**

In comparison with soil carbon, the results show much more consistency and convergent global number of carbon stock among the four global vegetation datasets (Figure 3). Our results show that global vegetation carbon stock is 10% to 25% of the global soil carbon stock, depending on soil depth. The high spatial correlations (r>0.75) between each pair of data indicate the current estimations of vegetation is consistent. Different from the spatial distribution of soil carbon, most vegetation carbon is located in the tropics whereas much less carbon in higher latitudes. As a matter of fact, the $C_{veg}$ in circumpolar region is only 10% of that in non-circumpolar region (Table 2). Here we address that the $C_{veg}$ is consist of three components including AGB, BGB and herbaceous biomass among which the herbaceous biomass is estimated from mean annual GPP (see Methods). We show the herbaceous biomass is only 5% of the total $C_{veg}$ and less than 1% of the total $C_{soil}$ indicating the minor role of herbaceous biomass in affecting the spatial distribution of total carbon stock and the uncertainty. The comparison among the four vegetation datasets shows relatively higher level of disagreement in arid and some cold regions (Figure 3, upper off-diagonal subplots). Nevertheless, the current estimations of global vegetation from different sources show consistent spatial distributions.

Our results show that the spatial pattern of the global GPP is similar to the $C_{veg}$ where there is higher primary productivity in the tropics and lower in the higher latitudes (Figure 4). The different members of the GPP estimation (see Methods) show very high consistency globally except for arid and polar region. The relative uncertainties in arid and polar region range from 50% to 100% whereas there is less than 50% of uncertainties in other regions.

Although the differences among different vegetation and GPP estimations, in general, are not as high as soil, we show the
460    uncertainties can be regionally high. We thus next investigate the contribution of each component to the spatial distribution
and uncertainty.

**4.5 The ecosystem carbon turnover times and associated uncertainties**

Using $C_{soil}$, $C_{veg}$ and GPP, we estimated the carbon turnover times with different combinations of datasets in order to
quantify the uncertainty. We calculated $\tau$ in the same manner as the previous study (Carvalhais et al., 2014) in which they
465    used only 1m of soil in the circumpolar region and full soil depth in the non-circumpolar region. We compared the spatial
distribution of the $\tau$ estimations of the previous and our study (Figure S5). Our results show a large range of relative
difference and low spatial correlation (r = 0.51). We found the main differences are in the northern circumpolar region,
which is caused by the differences in the $C_{soil}$ estimations, indicating a large uncertainty of $\tau$ estimations the in this region.
Our estimation of global mean $\tau$ is 35 years with an interquartile range of 29 to 43 years, which is much longer than the
470    previous study of 23 years with interquartile range of 19 to 30 years. In addition, we derived a global $\tau$ of 40 years with an
interquartile range of 29 to 47 years by assuming the maximum active layer thickness to be the full soil depth in the
circumpolar regions instead of using only 1-meter $C_{soil}$ as was done in the previous study. The incorporation of deep soil in
the circumpolar region increased the global $\tau$ by 7 years. The global spatial distribution of $\tau$ (Figure 5) shows great
heterogeneity, which ranges from 5 years in the tropics to over 1000 years in northern high latitudes. The results show a U-
475    shaped distribution of $\tau$ along latitudes where $\tau$ increases nearly three orders of magnitude from low to high latitudes. Figure
5b shows the map of relative uncertainty that is derived from different datasets. The higher relative uncertainty indicates
more spread among the datasets used to estimate $\tau$. Our result shows that peatland and arid regions generally have higher
uncertainties than the rest of the world. We found several regions with very different estimations of $\tau$ among the datasets
including north-east Canada, central Russia and central Australia where the relative uncertainties are over 100%.

$C_{soil}$, $C_{veg}$ and GPP contribute differently to the overall uncertainty of τ as shown in Figure 6. The difference among soil datasets is the dominating factor of τ uncertainty, especially in the circumpolar regions and the Indonesian peatland where there is large amount of soil organic carbon in subsoil. On the other hand, the uncertainties of τ in arid and semi-arid regions are controlled by the difference in GPP products. The contribution of vegetation to the uncertainty in τ is most significant in the tropics and warm temperate regions where there is large vegetation biomass. It is worth to note that contributions from each component also vary with depth of carbon stock that was used to calculate τ. For instance, the uncertainty contribution from $C_{veg}$ becomes smaller when the $C_{soil}$ up to 2 meters is used compared to only using 1-meter in calculating τ. However, the fact that the difference in the soil products was the major contributor to the τ uncertainty remains no matter what soil depth is used. Globally, the uncertainty of τ is mostly derived from soil and GPP, which dominate 82% and 17% of the global land area, while vegetation plays a minor role globally (1%).

**4.6 The zonal pattern of turnover times**

The latitudinal distributions of τ can be best represented by a second-degree polynomial function (Figure 7b). After fitting the data of all ensemble members, the rate of τ change with latitude can be obtained by taking the first derivative of the fitted polynomial function. We found that the rate of τ change (Figure 7c) has very consistent zonal patterns for different τ ensemble members from different data sources. The result shows a consensus on the change of τ with latitude of different datasets. We also found that the zonal τ gradients were not significantly ($P > 0.05$) different from each other for different selections of soil depth, indicating soil depth has no significant effect on the τ gradient along latitude. It is worth to note that there is a significant difference in the zonal τ gradient between the northern and southern hemisphere ($P < 0.0001$) and that τ increases faster from low to high latitude in northern latitudes than in the southern latitudes. The results show that we have high confidence in the zonal distribution of τ and that the difference across datasets does not affect the robustness of the pattern.

**4.7 The zonal correlation between turnover time and climate**

The correlations between τ and mean annual temperature and mean annual precipitation are analysed for all the ensemble members on global scale (see Method section). The correlation (Figure 8a) is the strongest in northern mid-to-high latitudes between 25º N and 60 º N, and it decreases rapidly from 20º N to the equator. In the southern hemisphere, it increases until 40º S, albeit having a weaker gradient than in the northern hemisphere. The uncertainties originating from different data sources are shown by the shaded area (Figure 8). The result shows that there are high uncertainties in the transitional regions between the temperate and Arctic regions (50 – 70º N) as well as tropical regions (20º N to 20º S). Similar to the previous result of uncertainty contribution where soil is the dominating factor, the differences in $C_{soil}$ also cause the spread in τ - T correlation. However, the patterns of correlation along latitude do not change regardless of the data source and the soil depth. All ensemble members agree that τ is negatively associated with temperature, with stronger associations in cold regions than in warm regions.

The correlation between $\tau$ and precipitation, in general, has larger variability across latitude and a higher uncertainty due to differences in data (Figure 8b). Contrary to the $\tau$ - T relationship, the uncertainty of the $\tau$ - P relationship derived from both different data sources and soil depths are smaller in the tropics than in high latitudes. Negative correlations dominate the latitudes between 20 and 50º N as well as between 20 and 40º S, while there is a stronger positive correlation in the tropics. There is a shift in the sign of the correlation coefficient from negative in temperate zone to positive in tropics, indicating the role of water changes from water-limited regions to water-excessive regions. We found the pattern of correlation between $\tau$ and precipitation is different from the previous study (Carvalhais et al., 2014), especially in the tropics. We therefore investigated the possible cause of the difference by mixing all components ($C_{soil}$, $C_{veg}$ and GPP) between the previous and current study. By examining the correlation between each mixed $\tau$ estimation and climate factors, our results show that positive correlation in the tropics is caused by the $C_{soil}$ (Figure S7). This is consistent with the previous results (Figure 1) which shows large difference in the spatial distribution of $C_{soil}$ in the tropics between the three soil datasets we used in this study and HWSD soil dataset.

**5 Discussion**

In this section, we will discuss the robustness of the current state-of-the-art estimation on global terrestrial carbon turnover times and their response to climate change. We first show the variation of spatial and vertical distribution of carbon stock in different regions and the possible reason for the difference, and we then discuss the robustness of zonal distribution of turnover times and zonal changing rates across different datasets. Finally, we focus on the sensitivity of turnover times to climate and implications.

**5.1 Estimation of global carbon stock**

Accurate estimation of terrestrial carbon storage and turnover time are essential for understanding carbon cycle-climate feedback (Saatchi et al., 2011; Jobbágy et al., 2000). Our analysis benchmarks soil, vegetation carbon storage and GPP from multiple state-of-the-art observational based datasets at global scale and provide not only an estimate of the total carbon stock but also the vertical distribution and spatial variability of global carbon stock. We divide the global map into circumpolar and non-circumpolar regions due to the different characteristics and uncertainty.

We found that there is a significant difference across the current soil carbon datasets in both circumpolar and non-circumpolar regions. The results show that the uncertainty of $C_{soil}$ estimations in the circumpolar region is two times larger than that of the non-circumpolar region (Figure 6). The spatial patterns of total ecosystem $C_{soil}$ among the soil datasets are more consistent in the non-circumpolar region than in the circumpolar region. In contrast with the non-circumpolar region, there is lower confidence in the circumpolar region in estimating $C_{soil}$ due the fact that there is low spatial correlation across datasets. The difference can be caused by various reasons. As an important input to the machine learning method, in-situ soil profiles are very important factors that influence the final results of the upscaling. The sparse coverage of soil profiles in the circumpolar region may cause the large divergence in the northern circumpolar region. A major difference between S2017

570   and the other two soil datasets is that soil carbon stock was a direct target of upscaling in the former dataset, while in the latter two datasets each component used to calculate $C_{soil}$ (carbon density, bulk density and percentage of coarse fragments) was predicted individually. In addition, the climatic covariates that were used in the upscaling were different (see Method).

In contrast with the non-circumpolar region, the circumpolar $C_{soil}$ does not have a decreasing trend up to 4 meters of soil depth (Figure S1) which indicates that there is a significant amount of carbon stores in deep soil. The deep soil turnover is a

575   key process to the global carbon cycle yet poorly understood (Todd-Brown et al., 2013). In this study, we extrapolated the soil carbon stock to full soil depth. We chose the model ensembles from a framework to pick out the models that had a minimum distance between prediction and observations by using in-situ soil profiles (see Supplement). Two model ensembles were selected that can best represent the soil vertical distribution in circumpolar and non-circumpolar regions by comparing model simulations and in-situ observations. The final results depend on the information from the soil profiles and

580   also the characteristics of the empirical models. The extrapolation gave us insights to the carbon storage and vertical distribution in deep soil. The results of extrapolation show there is approximately 15% of carbon stored below 2 meters globally and over 20% of carbon stored below 2 meters in the northern circumpolar region. Although the total amount of carbon storage in the ecosystem shows a large divergence among different datasets, the ratio between different soil depths are quite consistent indicating a high confidence in the vertical structure of soil compare to the total amount.

585   The global soil carbon stocks across observational-based datasets are much less divergent than the current earth system model (ESM) simulations. The CMIP5 results show the simulated carbon storage ranges from 500-3000 PgC making τ varies by a factor of 3.6, from 11 to 39 years (Todd-Brown et al., 2013). Our results show that the amount of carbon in the ecosystem is much higher than the estimation by ESMs. Even the lowest estimation (S2017 dataset) of total carbon storage is about 500 PgC higher than the highest ESM estimation (MPI-ESM-LR). The spatial distribution of carbon stocks among

590   ESMs have a large variation across models while the observational-based datasets are more consistent in the non-circumpolar region. But we leave a question mark to the soil carbon in the circumpolar region, which is characterized by large uncertainty as shown by the current observational-based soil datasets.

Compared with soil, the higher level of consistency in the vegetation and GPP estimates indicate there is a consensus on the current estimations in the above-ground carbon stock. However, we note that the regional differences in the products can

595   significantly affect the spatial distribution and uncertainty of τ. Nevertheless, vegetation and GPP contributes a little to the global mean value of τ estimations.

**5.2 The terrestrial carbon turnover time and uncertainty**

The uncertainty analysis showed that our current estimation of τ has a considerable spread which derived from the state-of-the-art observations of soil, vegetation and carbon fluxes. In this study, we showed the uncertainty is contributed mainly by

600   the soil carbon stock and GPP, where the former dominates the vast areas in the circumpolar region and the tropical peatland while the latter dominates the semi-arid and arid regions. We showed that GPP is the second largest contributor to the total uncertainty which potentially leads to significant differences in the estimation of τ considering different products of GPP (Zhang et al., 2017; Norton et al., 2019). This result is consistent with the previous study (Todd-Brown et al., 2013) that the

bias in estimated primary productivity can affect the carbon turnover estimations to a large extent not only by using observational-based data but in the ESMs simulations. However, the uncertainty comes not only from the differences across datasets but also from the soil depth we chose to estimate τ. The frozen permafrost soil in the circumpolar region, although containing a large amount of carbon is an important component in the process of turnover (Zimov et al., 2006). However, we do not know to what soil depth we should use in the τ estimation since currently our knowledge on the active layer thickness of frozen permafrost soil is still lacking. In addition, the active layer thickness of permafrost changes with climate, which adds more uncertainty to the estimation of τ. Thus, we argue that the current datasets cannot support robust estimation of global τ. It is worth to note that our estimation of τ is based on the steady-state assumption, that is, the net exchange of carbon between the terrestrial ecosystem and the atmosphere equals to zero. In our study, the steady-state assumption is a proper assumption for that our analysis focused on the τ estimation at long-term temporal scale and large spatial scale. However, this assumption is valid to a much less extent at site-level as the net exchange of carbon is, most of the time, not in balance (Ge et al., 2018).

Although the current estimation of τ has a large variation, we show that the zonal distribution of τ is a robust feature that changes little with different datasets, which indicates that the current state-of-the-art datasets all agree on the latitudinal gradient of the carbon turnover time. Another robust feature is that the zonal changing rate of τ does not change with the soil depth (Figure 7). It has always been a problem of what soil depth should we use to represent the functional part of carbon in the ecosystem. The selection of soil depth is usually arbitrary and varies from study to study. For example, Koven et al. (2017) and Wang et al. (2017) used the top one-meter of soil carbon to represent the total terrestrial carbon pool while Carvalhais et al. (2014) extrapolated soil to full depth and used it as the pool. Our results demonstrate that the selection of the soil depth does not affect the zonal pattern that we observed. This can be better seen in the next section with the response of τ to climate.

**5.3 Robust associations of τ and climate**

Despite the large uncertainty in the τ estimations, we identified robust response of τ to climate change. It is well recognized that the sensitivity of terrestrial carbon to climate is a major uncertainty, which is reflected by the spread of τ estimation by the different ESMs. However, we need reliable estimations of τ to quantify its climate sensitivity and provide robust constraints to improve the performance of the current ESMs. We showed the zonal correlation between τ and temperature varies with latitude where high correlations are found in the high latitude and low to moderate correlation in low latitude, especially the tropics. The zonal pattern of τ-precipitation is more complicated in that water availability can cause local variability to a great extent. The correlation between turnover times and precipitation in the tropics is higher than that with temperature as shown in Figure 8d indicating a potentially more dominant role of precipitation in the tropics. The role shifted along latitude between temperature and precipitation in the pattern of τ due to the variation in the relative importance for each parameter. However, the temperature gradient shaped the zonal distribution of τ as it can be seen that τ increases with latitude. All of these relationships are verified by each ensemble member of the data. We found the correlations, although they vary in strength, are very robust. The intimate interaction of energy and water along with other factors such as land use

change all affect τ but on different spatial and temporal scales. It is worth mentioning that the τ - T relationship is similar
when compared with previous results (Carvalhais et al., 2014) whereas there are considerable differences in the τ -
precipitation relationship, specifically in tropical regions where the turnover times were always negatively correlated with
precipitation in previous study. The different τ – P zonal patterns of correlation between the previous and the current study,
as shown before, is mainly caused by the difference in the soil carbon stock (Figure S7). This finding indicates the response
of τ to moisture is characterized by large uncertainty.

[revised manuscript text omitted]

**Supplementary material**

**1 Mass conservative aggregation**

In order to unify the spatial resolution and geographic coordinate system of dataset from different sources, we need to make sure that the total amount of stock for soil, vegetation, etc. doesn't change during aggregation and transformation, i.e. the variable need to be mass conservative. However, 'state' variable such as temperature, vegetation types do not need to fulfill the mass conservative requirement, nor they should. In our study, we developed a mass conservative method to maintain mass for carbon stocks. We first multiply the variable that need to be aggregated ($X_{fine}$) by corresponding land area ($A_{fine}$) at grid cell level represented by equation (1), then aggregate the product ($XA_{fine}$) by summing the values in N×N grids cell depending on the target resolution (equation (2)). The land area is also sum to the target resolution (equation (3)). Finally, the area-weighted variable is derived by dividing aggregated product ($XA_{coarse}$) by corresponding land area ($A_{coarse}$) as illustrated by equation (4).

$$XA_{fine} = X_{fine} \times A_{fine} \quad (1)$$

$$XA_{coarse} = SUM(XA_{fine}) \quad (2)$$

$$A_{coarse} = SUM(A_{fine}) \quad (3)$$

$$XA_{coarse} = \frac{XA_{coarse}}{A_{coarse}} \quad (4)$$

We applied the method to all datasets that requires aggregation including soil, vegetation and GPP that were used in the study.

**2 Bulk density correction**

The bulk density (BD) in SoilGrids and LandGIS are too high due to two reasons. First, the measurements of BD are less and missing in many horizons (Hengl et al., 2017). And the measurements of BD in permafrost region, especially in Canada forest soil and Russian, are problematic (personal communication with Tomislav Hengl). In this study, we applied a pedotransfer function from Köchy et al. (2015) to make correction based on organic carbon concentration (we only applied the function to the grid cells where carbon > 8%):

$$BD = (1.38 - 0.31 \times \log\left(\frac{OC}{10}\right)) \times 1000$$

925

**3 Model selection for extrapolation of soil**

In this section, we introduce the framework that we used to select the models for extrapolating soil from $0 - 2m$ to full soil depth.

**3.1 Different characteristics of permafrost and non-permafrost soil**

930 The amount and vertical distribution of soil organic carbon are largely influenced by vegetation which fixes atmospheric $CO_2$ and transport carbon into the land ecosystem. However, the SOC stock have a much more complicated relationship with productivity of plants than a simple linearly one (Jackson et al., 2017). The higher biomass, which implies more carbon sequestration by aboveground biomass, however, does not necessarily lead to increases in SOC storage. Although the processes of soil formation, accumulation, and stabilization have been intensively studied and debated, the mechanisms that

935 determine the soil carbon stock, especially in deeper soil, are still unclear. Instead of using process modelling approach, we chose statistical approach to extrapolate each soil profiles in the gridded dataset from 2m to full depth. The reason of performing soil carbon stock extrapolation is that we have little knowledge on how much the carbon stored in the soil that is deeper than 2m, although deeper soil is a crucial component in the climate-carbon cycle feedback. The other reason is the different dataset report SOC stock at different depths. The advantage of using statistical method is that we do not need to

940 know the mechanisms that control the soil processes. Instead, we select simple empirical mathematical models that can represent and predict the in-situ soil profiles.

We used 425 permafrost peatland profiles from ISCN soil database and 1000 profiles from WOSIS soil database to study the characteristics of vertical distribution of SOC. Figure S1 shows the accumulated SOC stock profiles with depths in permafrost and non-permafrost region. The vertical distribution of carbon with depth in permafrost soil has a distinguished

945 feature that the SOC has a high linear relationship with depth. This fact implies the soil carbon keeps increasing even after 3 meters in permafrost soil (Figure S1b). However, we have no idea to what depth can soil carbon keep increasing and the total amount of the storage in permafrost peatland due to the limited observational depth of SOC. In contrast, soil profiles in non-permafrost region stop increasing mostly before 2 meters. The results demonstrate the necessity of extrapolating soil to full depth, especially for permafrost soil.

950 ### 2.2 Selection of models

We included 12 models (Table S1) for predicting SOC stock to full soil depth. Figure S2 shows an example result in which the data points that is shallower than 1m were used to fit all the models and predict the point that is deeper than 2m for a

typical soil profile. Due to the different mathematical characteristics of the models, the prediction has quite a spread. Relatively 'conservative' models including model ensemble BHIJKL tend to underestimate the carbon stock while the more 'aggressive' ones ACDEFG tend to overestimate the stock.

955

In the sense that we do not know which one or group of models can best predict the accumulated carbon storage, we conducted a selection process (see Methods) by grouping all the models into all possible combination and rank the performance for all the model averaging results as shown in Table 2. All the models were used to fit the WOSIS data which covers most of the biomes and ISCN database which covers only permafrost soil. We conducted three batch of experiments

960

in the same manner but used data points within different depths. The data points lower than 50cm, 100cm and 200cm were used to predict the SOC that is higher than 200cm. Our goal is to find the ensemble of models that has the highest model performance, the best coverage, the minimum error and AIC.

**2.3 Extrapolating soil with different method**

965

The main goal of using several models is to search for the best group of models that can best predict the vertical distribution of soil carbon stock and we compared the below approaches for that purpose:

1. The Bayesian Model Averaging (BMA) method is used in this study to find the best model ensemble for the prediction of soil carbon storage to full depth. The MODELAVG Matlab toolbox (Vrugt, 2016) which implemented many different model averaging techniques including BMA method. The advantage of BMA method is that it considers explicitly the

970

   uncertainty of prediction of a target variable which can provide a probabilistic distribution of weight for each model instead of only a weighted-average, deterministic prediction. By maximize the likelihood function from the training dataset, the weights $\beta = \{\beta_1, …, \beta_k\}$ and standard deviation $\sigma = \{\sigma_1, …, \sigma_k\}$ are estimated.

2. Equal weights averaging (EWA) which consider each the participating model have the same weight and the prediction is derived by equal-weighted averaging the model results.

975

The complete combination among different models are also compared and the best model ensembles are obtained by maximizes MEF, minimizes KL and minimizes AIC. The results show that EWA and BMA methods have similar performances (Table S2). We choose EWA method due to it have a slightly better coverage of observations.

Two model ensembles were selected from the model selection framework that can best represent circumpolar and non-

980

circumpolar region based on observational datasets in the two regions, respectively. The performance of the chosen ensemble is synthesized in Figure S3. It shows the ensemble DIJKL overall can well predict the carbon stock in non-circumpolar region that is deeper than 200cm only using points lower than 50cm. Model efficiency is 0.83 and the residue between observation and prediction is little biasd in the prediction (Figure S3c). The histogram (Figure S3b) shows the prediction has the same distribution as the observations. The ensemble was also used to predict observations within different

985

percentiles and over 70% of observations can be included in the uncertainty ([$-\sigma$, $+\sigma$]). The results show that the selected

models can well represent the vertical distribution of $C_{soil}$ thus we used them to extrapolate the global gridded datasets in order to obtain the total soil carbon storage in the soil. The selected model ensemble ACDEF for the circumpolar soil have lower model efficiency and less well represent the soil in the region (Figure S4). We then applied extrapolation on three global datasets which are Sanderman, SoilGrids and LandGIS. The averaged results of ensemble DIJKL is used to extrapolate non-circumpolar soil from 2m to full soil depth and ensemble ACDEF to extrapolate circumpolar soil.

[Figure]

Figure S9: **The vertical distribution of accumulated SOC stock (kg.m-2) with depth (cm).** (a) 425 soil profiles of permafrost peatland region and (c)1000 soil profiles of non-permafrost region. The probability distribution density of SOC for (b) permafrost, (d) non-permafrost. The blue open circle represents observational data points in each profile.

1005

1010

[Figure]

1015    Figure S10: **An example of soil profile vs models.** Overlay the observational points and model results.

[Figure]

1020   **Figure S11: Performance of the averaged results of model D, I, J, K and L in predicting soil carbon storage from 50cm to 200cm using WOSIS data.** (a) Ensemble mean vs. observation, 1:1 line in blue. (b) The histogram of observation, model ensemble and each model. It shows the Kullback-Leibler distance from model ensemble mean to observation, the two-sample Kolmogorov-Smirnov test (1 represent the model ensemble mean and the observation come from the same distribution, 0 otherwise), the p-value of Kruskal-Wallis test (significant if p<0.05). (c) residue between model ensemble mean and observation. KS represents the one-sample Kolmogorov-Smirnov
1025   test (1 represent the model ensemble mean and the observation come from the same distribution, 0 otherwise). AD represents Anderson-Darling test (1 represent the model ensemble mean and the observation come from the same distribution, 0 otherwise). (d) The coverage of observation data points within [-σ, +σ], [min, max], [25%, 75%] and average.

[Figure]

1030 **Figure S12: The same as Figure 3 except for using ISCN data and model ensemble of A, C, D, E and F to predict soil carbon storage from 200cm to deep soil.**

[Figure]

1035 **Figure S13: Comparison of τ estimations between the previous study (Carvarhais et al., 2014) and the current study.** The upper off-diagonal subplot is the ratios between each pair of datasets (column/row). The bottom off-diagonal subplot shows the major axis regression between each pair of datasets (m: slope, b: intercept, r: correlation coefficient).

[Figure]

1040 **Figure S14: Global distribution of full soil depth.**

[Figure]

**Figure S15: The zonal pattern of the correlation between τ and climate factors.** Each component ($C_{soil}$, $C_{veg}$ and GPP) from the previous study (Carvalhais et at., 2014) is mixed with each component of the current study. The prefix 'old' stands for the component from the previous study and the prefix 'new' stands for the component from the current study.

1045

**Table S1:** Empirical functions candidates for extrapolation of soil carbon

| | Equation |
|---|---|
| A | $a \cdot D^b + C$ |
| B | $a \cdot e^{b \cdot D} + c \cdot e^{d \cdot D}$ |
| C | $a \cdot \log(b \cdot D + 1)$ |
| D | $a \cdot \log(b \cdot D + c)$ |
| E | $K \cdot \log_{10}(D) + I$ |
| F | $(10^I \cdot D^{K+1})/(K+1) + c$ |
| G | $a + b \cdot D$ |
| H | $b \cdot (1 - \beta^D)$ |
| I | $b \cdot (1 - \beta^D)^a$ |
| J | $a \cdot (1 - e^{-(D/b)^c})$ |
| K | $a \cdot (1 - e^{-b \cdot D})^c$ |
| L | $a \cdot (1 - \dfrac{\log(1 - (1-b) \cdot e^{-c \cdot D})}{\log(b)})$ |

1050

1055

**Table S2: Performance of different methods.**

| | Circumpolar | | Non-circumpolar | |
|---|---|---|---|---|
| | **EWA** | **BMA** | **EWA** | **BMA** |
| **RMSE** | 36.575 | 37.686 | 5.482 | 5.292 |
| **AIC** | 1516.134 | 1528.526 | 3977.226 | 3895.513 |
| **KL** | 0.020 | 0.020 | 0.039 | 0.036 |
| **MEF** | 0.640 | 0.617 | 0.862 | 0.872 |
| **Coverage (%)** | 14.5 | 13.5 | 61.9 | 50.3 |

---

## Author Response (AR1)

**Response to the Referee's comments**

**Dear Referees:**

- 5 We have now completed the revision of the manuscript "Apparent ecosystem carbon turnover time: uncertainties and robust features" as per the suggestions and comments from the Reviewers. This final revision took longer than we expected, especially due to the deeper investigation needed to address the questions and suggestions made by Reviewer #1. The major changes include:
- A deeper analysis on the role of uncertainties in vegetation carbon to the uncertainties in turnover times;
  - Reshaping the manuscript for a better balance on the contributions of GPP and Cveg to the estimation of turnover times, in comparison to the more prominent focus of Csoil in the original manuscript. To do that, we added new analysis of the data (independent GPP estimate) and included new points into the discussion sections.

**15**

- Discussing relevant aspects related to the steady-state assumption and perspectives related to model comparisons, especially ESMs.
- Clearing any potential sources of confusion or lack of clarity.
- Major editorial revisions for consistency and clarity of the text throughout.
- 20 We feel that the revised manuscript fully addresses both of the Reviewers' concerns and makes the manuscript clearer and more comprehensive. Thanks to the comments, the revised manuscript is now a critical appraisal of this new dataset. We would appreciate any advice if any of the revisions could be improved, and look forward to doing so, if and when needed. We are deeply grateful for all the constructive comments and editing recommendations and look forward to moving this process forward

25 in a positive way.

Please accept our kind regards,

Naixin Fan and Nuno Carvalhais, on behalf of the co-authors

**Notation:**

30 Sentences in bold black color are the original comments from the Reviewer and our responses are marked in blue color; the specific changes made in the manuscript, where appropriate, are transcribed after our answers (in italic) and the line numbers are indicated. LR stands for the Line number in the revised version of the manuscript, also please see the marked-up manuscript attached after the responses to the referees:

1

**Response to Referee #1**

I would be interested to hear more information about some of the derived datasets. For example, the

- 40 creation of the herbaceous carbon stock map is described but what is the relative proportion of vegetation carbon found within the herbaceous layer is not stated? As the GPP ensemble is used in the estimation of the herbaceous layer what is the uncertainty in the herbaceous carbon content? How does the herbaceous carbon stock influence ecosystem turnover time vary in space, i.e. could it have been neglected?
- 45 This is a pertinent question and thank you for bringing it up. Overall, the herbaceous biomass plays only a minor role in the estimation of  $\tau$  since it is less than 1% of soil carbon stock and less than 5% of vegetation carbon stock. Thus, it has minor contributions to the global estimation of carbon turnover times. Also, the spatial correlation between the different Cveg estimates, with and without the herbaceous components, is high (Figure 3 vs. Figure S7, the latter was newly added to address this issue), and
- 50 locally these differences are marginal. The local effects on  $\tau$  are dependent on the contribution of the Cveg term itself to the total ecosystem carbon, although these local differences will not be higher than 1 year (in the extreme case that the herbaceous mass equals to GPP). As such, the herbaceous carbon stocks show a negligible effect in changing the global and local estimates of whole ecosystem carbon turnover times (Figure S7). We also add now Figure S7 to the manuscript.

**55**

- Figure S7: The spatial distribution of turnover times with different estimates of herbaceous carbon stocks. The turnover times are estimated with no herbaceous component ("Herb (non)"), herbaceous components based on the different percentiles of GPP estimates: 25% ("Herb (50%)"), 50% ("Herb (50%)"), 75% ("Herb (75%)"). The global turnover times are shown in the bottom of each diagonal subplot. The upper off-diagonal subplots are the ratios between each pair of datasets (column/row). The bottom off-diagonal subplots
- 60 show the major axis regression between each pair of datasets (in: slope, b: intercent, r: Pearson correlation coefficient). The ranges of both of the colorbars approximately span between the 1st and the 99th percentiles of the data. Hereafter, all figures comparing different spatial maps include the information in a similar manner.

We have now included this analysis in the updated version of the manuscript (LR352-358), where we 65 can now read:

The  $C_{veg}$  consists of three components including AGB, BGB and herbaceous biomass. The herbaceous biomass is estimated from mean annual GPP (see Methods 3.2, Carvalhais et al., 2014), and globally represents 5% of the total  $C_{veg}$  and less than 1% of the total  $C_{soil}$ , indicating a minor role of herbaceous biomass in affecting the global estimates and the spatial distribution of  $\tau$ . The comparison among the

70 four vegetation datasets shows a mean of 410 PgC in  $C_{veg}$ , with a spread of 11% across the different datasets, and a consistent spatial distribution across the different sources. Locally these differences can be higher, as observed in the relatively higher level of disagreement in sparse vegetated arid and some cold regions (Figure 3, upper off-diagonal subplots).

75 Similarly, the soil carbon estimated to maximum depth would be interesting to investigate further. A really simple but nice addition would be a map of the maximum soil depths inferred by your analysis.

We added the soil depth global distribution map (Figure S5). Please also note that the full soil depth is not inferred by our analysis but is provided by a global dataset (Webb et al., 2000, in Methods section too).

80

95

Figure S5: Global distribution of full soil depth according to Webb (2000).

The current text is a little unbalanced towards  $C_{soil}$  sometimes to the exclusion of  $C_{veg}$  or GPP in the introduction, results and discussion sections. The introduction sets out the overall challenge and

- 85 usefulness of such datasets in constraining Earth System Models and their role in quantifying the response of the terrestrial ecosystem to climate change. However, the fact that this is an update paper is not made fully clear. Doing so would I think make it straight forward to highlight the weaknesses of the previous analysis and how they are being improved here making a more robust and unique dataset. I honestly do support making updates and improvement to existing datasets as this provides a clear
- 90 traceable advancement in the science. Because the current manuscript does not clearly highlight soil as a weakness / uncertainty of existing works the introduction reads as being very soil dominated with little introduction of the vegetation carbon stock challenges or the estimation of GPP.

Yes, that's correct. There is a stronger balance towards  $C_{soil}$  as it is the dataset holding the heaviest uncertainties. We make this clear in the introduction text now. We also agree that this unbalance is too detrimental to the importance of  $C_{veg}$  and GPP to the estimates of  $\tau$  and have introduced several analysis

elements and a discussion section in the updated version of the manuscript to have them in a more

balanced way. The distribution of  $C_{veg}$  and GPP are added as two new paragraphs in LR 343-365. Please see here the transcript of the added sections:

**4.3 The spatial distribution of vegetation**

100 Different from the spatial distribution of soil carbon, most vegetation carbon is located in the tropics whereas much less carbon in higher latitudes. In fact, the Cveg in circumpolar region is only 10% of that in non-circumpolar region (Table 2).

[revised manuscript text omitted]

The introduction does clearly state one of the key assumptions, that ecosystems are assumed to be in steady state. What is missing is an appreciation that much of the worlds vegetation is not in steady state, either due to direct human intervention (biomass removal or other land use change) or as a result of increasing CO2 concentration. Attempting to quantify this is out of scope but I think it would be useful to include either in the introduction or discussion the potential implications of this assumption leading

160 to include either in the introduction or discussion the potential implications of this assumption to an underestimate in turnover times (e.g. Ge et al., 2018).

Yes, we agree with the perspective of the Reviewer and have added a discussion to the manuscript to address this. In LR 507-514 we added:

It is worth noting that here the estimation of τ is based on the steady-state assumption, that is, the
assumption of a balance net exchange of carbon between terrestrial ecosystems and the atmosphere.
Here, the assumption is that integrating at larger spatial scales, by averaging the local variations in sink and source conditions, reduces the differences between assimilation and out-fluxes relative to the gross influx; and that the integration of stocks and fluxes for long time spans reduces the effects of transient changes in climate and of inter-annual variability in τ estimates. However, this assumption is
valid to a much less extent at smaller spatial scales (site-level) and shorter time intervals, as the

ecosystem-atmosphere exchange of carbon is most of the time not in balance and forced steady state

assumptions can lead to biases in estimates of turnover times and other ecosystem parameters (Ge et al., 2018; Carvalhais et al, 2008).

The results section, like the introduction, seems to be biased towards soil carbon results rather than a complete overview. This should be addressed. Further information can be found below in the technical comments. The discussion lacks any discussion of the vegetation carbon stocks and almost any discussion of the GPP estimates. I also find it odd that figures 1-4 are not mentioned in the discussion at all.

Yes, like mentioned above by the Reviewer too. We agree and we trust that with the addition of analysis
 and discussion on the Cveg and GPP components in the updated manuscript, as described above, is
 showing a more balanced contribution of the different components. We also added Figures 3 and 4
 comparing the different vegetation and GPP products; and made sure that Figures 1-4 are mentioned in
 the discussion (Section 5.1). Please see here the transcript of the Section 5 (LR421-468):

**5.1 Estimation of global soil carbon stocks**

[revised manuscript text omitted]

---

## Author Response (AR2)

**Response to the Topical Editor's comments**

Dear Topical Editor:

We thank you for pointing out these final editing issues that have escaped our attention before. We have addressed them all as suggested. Please see in detail below.

We would like to thank you also for the work and the attention given to this manuscript.

Meanwhile, we would also like to request you to replace the "Responses to Reviewers" in the interactive discussion page. The ones currently in the open discussion were updated with the details from the last submitted version of the manuscript, and do not fully reflect the final changes in the revised manuscript. The "Author Responses" files were also part of the previous submission of revision, or we can gladly upload/send the latest version given your kind response.

With our best regards,
Naixin Fan and Nuno Carvalhais, on behalf of the co-authors

**Notation:**
**Sentences in bold black color are the original comments from the Topical Editor and our responses are marked in blue color; the specific changes made in the manuscript, where appropriate, are transcribed after our answers (in italic) and the line numbers are indicated. LR stands for the Line number in the revised version of the manuscript:**

1. **ESSD strongly opposes registration barriers. Please justify! These seem especially in violation of German open data principles and in contrast to other European data repositories.**

Response: Even though the data is located inside the registration barrier, it is still publicly available to everyone interested in using it for research. We have the registration process for the purpose of keeping the list of contacts of users of the data. This list is available only to the contact person within the department and is used solely to notify all the users when there are unforeseen changes in the data (say, correcting for format, fixes in bugs, etc.), even though such changes are rare. Note that the list is not shared with other groups or institutions, nor it is used for any promotional activities, which is in line with the European data privacy policies. Finally, we would like to note that the BGC data portal, with similar registration process, has been extensively used to publicly distribute and update several datasets that have been produced by the department (for example, the large ensemble of FLUXCOM at http://fluxcom.org, and several products from the EU-funded H2020 BACI project at http://baci-h2020.eu/index.php/Projects/Projects

**2. Some text missing or confused at lines 344, 345?**

Response: We double checked and there are no missing texts in the referred lines. However, we found the sentence is confusing which may lead to misunderstanding. We thus revise the previous sentence to: "*Different from the spatial distribution of soil carbon, most vegetation carbon is stored in the tropics whereas much less carbon resides in the higher latitudes. In fact, the $C_{veg}$ in the circumpolar region is only 10% of that in the non-circumpolar region (Table 2).*" LR344-346

**3. Something missing at lines 433, 434?**

Response: We double checked and there are no missing texts in the referred lines. But we found an unnecessary reference to the Methods section remained due to editorial mistake, which is indeed confusing. The corrected sentence now is: "*Additional discrepancies may also be associated with the differences in climatic and other input covariates used in the upscaling which may yield a different estimation of $C_{soil}$ (see Section 2.1).*" LR434-434

**4. Line 440: SOC (soil organic carbon) acronym introduced here without prior definition?**

Response: That is correct. Thank you for pointing it out. We now introduce the meaning of the acronym in LR440: "*…for more robust predictions of soil organic carbon (SOC) with depth…*".